# Engineering triangular carbon quantum dots with unprecedented narrow bandwidth emission for multicolored LEDs

Fanglong Yuan[1], Ting Yuan[1], Laizhi Sui[2,3], Zhibin Wang[4], Zifan Xi[1], Yunchao Li [1], Xiaohong Li[1], Louzhen Fan[1], Zhan'ao Tan[4,5], Anmin Chen[2], Mingxing Jin[2] & Shihe Yang[6]

Carbon quantum dots (CQDs) have emerged as promising materials for optoelectronic applications on account of carbon's intrinsic merits of high stability, low cost, and environment-friendliness. However, the CQDs usually give broad emission with full width at half maximum exceeding 80 nm, which fundamentally limit their display applications. Here we demonstrate multicolored narrow bandwidth emission (full width at half maximum of 30 nm) from triangular CQDs with a quantum yield up to 54–72%. Detailed structural and optical characterizations together with theoretical calculations reveal that the molecular purity and crystalline perfection of the triangular CQDs are key to the high color-purity. Moreover, multicolored light-emitting diodes based on these CQDs display good stability, high color-purity, and high-performance with maximum luminance of 1882–4762 cd m$^{-2}$ and current efficiency of 1.22–5.11 cd A$^{-1}$. This work will set the stage for developing next-generation high-performance CQDs-based light-emitting diodes.

[1] College of Chemistry, Beijing Normal University, 100875 Beijing, China. [2] Institute of Atomic and Molecular Physics, Jilin University, 130012 Changchun, China. [3] State key Laboratory of Molecular Reaction Dynamics, Dalian Institute of Chemical Physics, Chinese Academy of Sciences, 457 Zhongshan Road, 116023 Dalian, China. [4] State Key Laboratory of Alternate Electrical Power System with Renewable Energy Sources, School of Renewable Energy, North China Electric Power University, 102206 Beijing, China. [5] Beijing Advanced Innovation Center for Soft Matter Science and Engineering, Beijing University of Chemical Technology, 100029 Beijing, China. [6] Guangdong Key Lab of Nano-Micro Material Research, School of Chemical Biology and Biotechnology, Shenzhen Graduate School, Peking University, 518055 Shenzhen, China. Correspondence and requests for materials should be addressed to L.F. (email: lzfan@bnu.edu.cn) or to Z.a.T. (email: tanzhanao@mail.buct.edu.cn) or to S.Y. (email: yangsh@pkusz.edu.cn)

Getting light from carbon has long been a dream of the scientific community. Carbon-based highly efficient light-emitting materials will not only raise the prospect of the next-generation technological frontiers of carbon photonics and optoelectronics, but also offer an alternative to traditional semiconductor inorganic quantum dots (QDs) such as the $Cd^{2+}/Pb^{2+}$-based QDs in light-emitting applications taking advantage of carbon's high stability, low cost, high abundance and environment-friendliness[1–7]. The discovery of room temperature light emission from quantum sized carbon (<10 nm) in 2006[8] triggered intensive researches on light-emitting carbon quantum dots (CQDs) to realize their wide potential applications[9–14]. In the last few years, great progress has been made in designing and synthesizing highly tunable bandgap fluorescent CQDs with a quantum yield (QY) as high as 75%, even comparable to the best performing $Cd^{2+}/Pb^{2+}$-based QDs, through a variety of strategies such as heteroatom doping, surface engineering or passivation, and product separation and purification[15–18]. Meanwhile, the wide potential optoelectronic applications of the CQDs have also been demonstrated[17–23]. For instance, light-emitting diodes (LEDs) from blue to red based on the bandgap fluorescent CQDs have been reported most recently[18], laying a solid foundation for the development of a new display technology based on the CQDs. However, despite the intensive work on the electronic and optical properties of CQDs, it has until now remained a widely accepted belief that CQDs can only give broad emission and inferior color-purity with full width at half maximum (FWHM) commonly exceeding 80 nm[15–19,23]. Indeed, such a broad bandwidth emission is far inferior to that of $Cd^{2+}/Pb^{2+}$-based QDs (FWHM <40 nm), which has severely hindered the application of CQDs-based LEDs in high color-purity displays[24,25].

The exact mechanism of the broadband fluorescence spectra of CQDs has been a longstanding unsettled issue. It has been generally believed that the broadband fluorescence spectra of CQDs stem from a broad size distribution. However, even after narrowing the size distribution through elaborate separation and purification, the fluorescence spectra still remains broad (FWHM >80 nm)[15–19,26]. This indicates that the broadband fluorescence spectra cannot be simply ascribed to the dot size polydispersity, but may be intrinsic to the CQDs. For instance, the complex nonradiative excited-state relaxation processes arising from specific structure-associated phenomena, such as self-localized charges and surface defect-trapped carriers observed in traditional inorganic QDs, are probably the main origin of the broadband fluorescence spectra of CQDs[1–3]. The former can be considered as transient defects formed in the excited states where photogenerated charge carriers are stabilized through large-amplitude vibrations and distortions driven by strong electron–phonon coupling[27–29], while the latter is usually induced by the numerous electron-withdrawing oxygen-containing groups such as carboxyl, carbonyl, and epoxy groups at edge or basal plane sites of the CQDs[26,30–33]. Consequently, weakening the electron–phonon coupling and reducing the surface defects by structural engineering may be a feasible way to realize narrow bandwidth emission of CQDs with high color-purity.

Resorting to common sense, the fact that triangle is the most stable structure in nature offers a hint in the structural engineering along the direction. Indeed, the physical properties including band-gap renormalization, electron–hole attraction, oscillator strength, and exciton polarization of model triangular graphene quantum dots (T-GQDs) containing 168 and 132 $sp^2$-hybridized C atoms have been theoretically investigated[34]. Shortly thereafter, the biexciton binding of Dirac fermions of T-GQDs containing 168 $sp^2$-hybridized C atoms has also been probed by transient absorption measurements and microscopic theory[35].

Here, we report the synthesis of high color-purity, narrow bandwidth (FWHM of 29–30 nm), and multicolored (from blue to red) emission triangular CQDs (T-CQDs) with a quantum yield up to 54–72%. The synthesis is conducted by judiciously choosing a three-fold symmetric phloroglucinol (PG) as the reagent (a triangulogen) together with a tri-molecular reaction route designed into the neighboring active -OH and -H groups for six-membered ring cyclization, propagating to the target high color-purity narrow bandwidth emission T-CQDs (NBE-T-CQDs) (Fig. 1a–e). The triangular structure and the narrow bandwidth emission of NBE-T-CQDs have been rigorously established, and their correlation manifest that the triangular structural rigidity dramatically reduces electron-phonon coupling, giving rise to the free-excitonic emission with negligible trap states. This has been borne out by elaborate theoretical calculations, which show highly delocalized charges and high structural stability of the T-CQDs. The multicolored LEDs based on the NBE-T-CQDs display high color-purity (FWHM of 30 nm) and high-performance with a maximum luminance ($L_{max}$) of 4762 cd m$^{-2}$ and current efficiency ($\eta_c$) of 5.11 cd A$^{-1}$. Moreover, the LEDs demonstrate outstanding stability both on shelf and in operation.

## Results

**Synthesis of NBE-T-CQDs.** Synthesis of the NBE-T-CQDs, as shown in Fig. 1a, involves the solvothermal treatment of three-fold symmetric PG triangulogen at 200 °C with different reaction time, followed by purifying via silica column chromatography using a mixture of dichloromethane and methanol as the eluent. The starting material PG triangulogen possesses a unique structure with three highly reactive hydrogen atoms at the three meta-positions activated by three electron-donating hydroxyl groups in a single molecule, which is a key point for the synthesis of the NBE-T-CQDs. For tuning their emission color, an appropriate amount of concentrated sulfuric acid was added as catalyst in the ethanol solution to control the size of NBE-T-CQDs (see Methods for more details). The preparation yield for NBE-T-CQDs is estimated to be about 8–13%. The typical aberration-corrected high-angle annular dark-field scanning transmission electron microscopy (HAADF-STEM) images of the NBE-T-CQDs (Fig. 1b–e) clearly demonstrate the almost defect-free graphene crystalline structure with an obvious triangular shape. To the best of our knowledge, this is the first time that the exquisite aberration-corrected HAADF-STEM images of carbon materials were obtained. The bright multicolored emissions of blue (B), green (G), yellow (Y), and red (R) were observed from the NBE-T-CQDs solutions with gradually increasing sizes from 1.9 nm, to 2.4, 3.0, and 3.9 nm, respectively (Fig. 1f), as expected from the quantum confinement effect[7,16–18]. Significantly, the tunable emission colors from blue to red could be observed even under daylight excitation (Fig. 1f), which is a clear signature of the strong emission. And the emission colors are brighter under UV light irradiation (excited at 365 nm, Fig. 1g) due to the pronounced absorption of NBE-T-CQDs at 365 nm (Supplementary Figs. 1-2).

**Optical properties of NBE-T-CQDs.** The most important and distinctive features of the NBE-T-CQDs that set them apart from all other previous reported CQDs are the extremely narrow excitonic absorption and emission peaks. Figure 1h shows the absorption spectra of the NBE-T-CQDs with strong and narrow excitonic absorption peaks centered at 460 (B-), 498 (G-), 521 (Y-), and 582 nm (R-NBE-T-CQDs), which is quite different from that of the previous reported CQDs with ultrabroad absorption bands[7–18,23,26]. The fluorescence spectra of NBE-T-CQDs

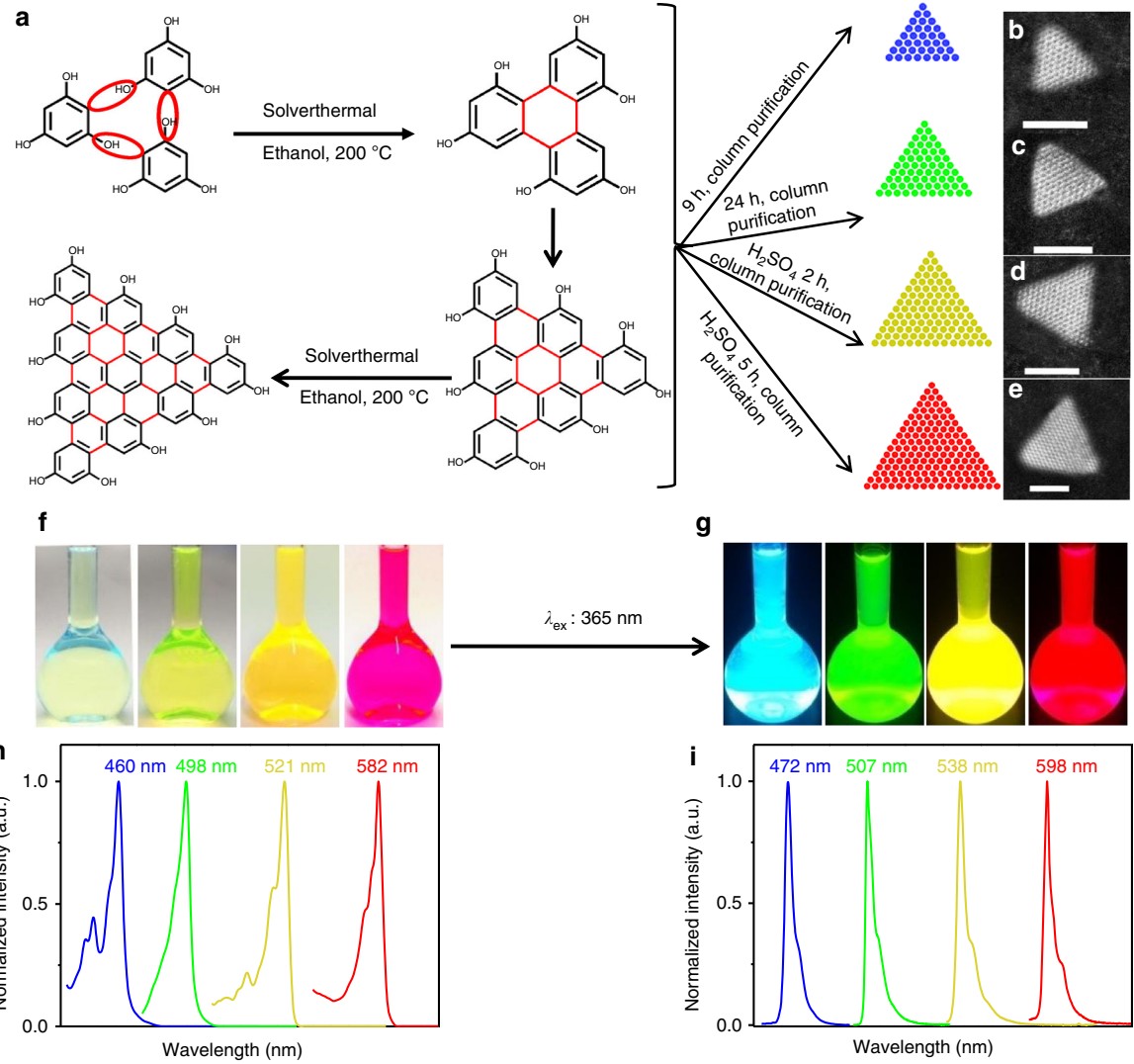

**Fig. 1** Design and synthesis of narrow bandwidth emission triangular CQDs. **a** Synthesis route of the NBE-T-CQDs by solvothermal treatment of PG triangulogen. The typical aberration-corrected HAADF-STEM images of B- (**b**), G- (**c**), Y- (**d**), and R-NBE-T-CQDs (**e**), respectively. Scale bar, 2 nm. Photographs of the NBE-T-CQDs ethanol solution under daylight (**f**) and fluorescence images under UV light (excited at 365 nm) (**g**). The normalized UV-vis absorption (**h**) and PL (**i**) spectra of B-, G-, Y-, and R-NBE-T-CQDs, respectively

(Fig. 1i) also show sharp excitonic emission peaks centered at 472 (B-), 507 (G-), 538 (Y-), and 598 nm (R-NBE-T-CQDs) with extremely narrow FWHM values of only 30, 29, 30, and 30 nm, respectively, which is far superior to previous reported CQDs with broadband fluorescence spectra (FWHM >80 nm) and even superior to the best $Cd^{2+}/Pb^{2+}$-based QDs (FWHM <40 nm)[7–18,23,26]. The weak shoulder emission peaks at longer wavelengths in the photoluminescence (PL) spectra may be ascribed to the excimer emission of the NBE-T-CQDs, which is often observed in highly delocalized polyaromatic systems[36,37]. Moreover, the NBE-T-CQDs also exhibit ultrasmall Stokes shifts of 12, 9, 17, and 16 nm for the B-, G-, Y-, and R-NBE-T-CQDs, respectively (Supplementary Figs. 1-2), much smaller than those of the common CQDs (Stokes shifts >80 nm)[15–18], implying the band edge direct exciton recombination of the optical transitions as well as the weak electron–phonon coupling of the NBE-T-CQDs. The maximum peak wavelength of the FL excitation spectra are centered at about 460 (B-), 498 (G-), 521 (Y-), and 582 nm (R-NBE-T-CQDs), and agree well with the corresponding excitonic absorption peak wavelengths (Supplementary Figs. 3-4), clearly suggesting that the emission of NBE-T-CQDs originates

from band-edge exciton-state decay rather than from defect-state decay. The unusually narrow emission peaks of the NBE-T-CQDs are irrespective of the wide excitation wavelength range (Supplementary Fig. 5). These results further confirm that the PL emissions of the NBE-T-CQDs originate from direct exciton recombination. This is very different from the traditional CQDs whose excitation-dependent fluorescence is dominated by surface defects[15–18]. The gradually red-shifted narrow bandwidth excitonic emission peak of the NBE-T-CQDs from 472 nm (blue) to 598 nm (red) are well consistent with the corresponding increased size from 1.9 to 3.9 nm. Significantly, such a correlation is almost linear as shown in Supplementary Fig. 6, a very clear characteristic of the bandgap transitions of the NBE-T-CQDs.

To gain more insight into the exciton recombination dynamics, we measured time-resolved PL spectra at emission and excitation wavelengths of about 472/460, 507/500, 538/520, and 598/580 nm for B-, G-, Y-, and R-NBE-T-CQDs, respectively, and the results are shown in Fig. 2a, which evidence monoexponential decay with fluorescence lifetimes of about 7.3, 8.3, 7.0, and 6.6 ns for the B-, G-, Y-, and R- NBE-T-CQDs, respectively (Supplementary Fig. 7). The monoexponential decay characteristics indicate that

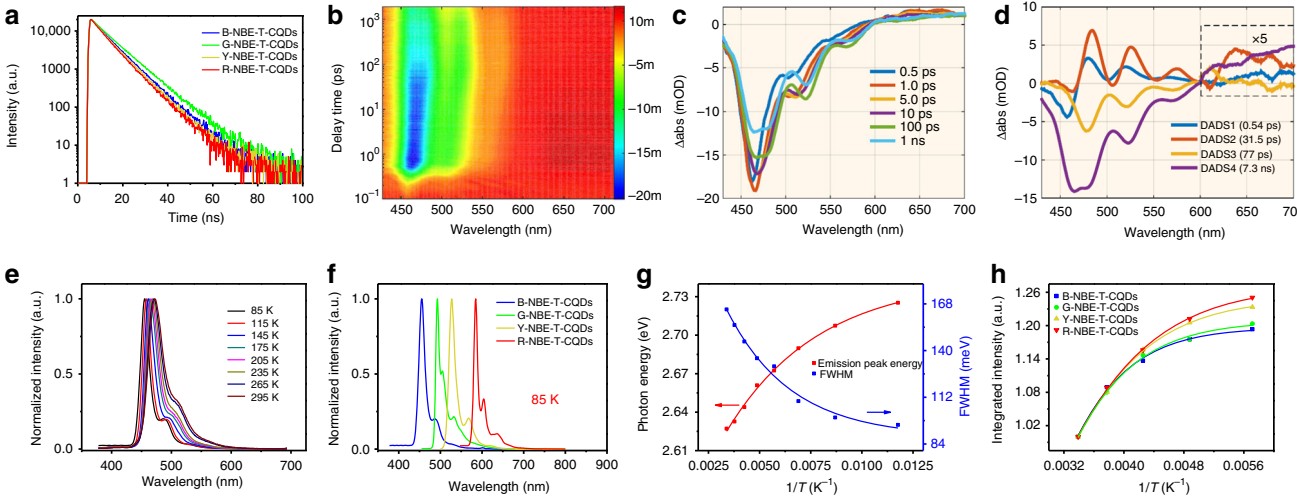

**Fig. 2** Ultrafast dynamics of the photoexcited states and temperature-dependent PL spectra of the NBE-T-CQDs. **a** Time-resolved PL spectra of the NBE-T-CQDs. **b** Two-dimensional pseudocolor map of TA spectra of B-NBE-T-CQDs expressed in ΔOD (the change of the absorption intensity of the sample after excitation) as a function of both delay time and probe wavelength excited at 400 nm. **c** TA spectra of B-NBE-T-CQDs at indicated delay times from 0.5 ps to 1 ns. **d** Results of the global fitting with four exponent decay functions. **e** The normalized temperature-dependent PL spectra of B-NBE-T-CQDs. **f** The normalized PL spectra of the NBE-T-CQDs acquired at 85 K. **g** The plots of the emission peak energy and FWHM of B-NBE-T-CQDs as a function of temperature (85–295 K). **h** The plots of integrated PL emission intensity of the NBE-T-CQDs as a function of temperature (175–295 K)

the excitons are highly stable and the radiative decay is extremely pure with a minimal nonradiative contribution[17,18], which is conducive to efficient fluorescence emission and again strikingly different from those reported CQDs with multi-exponential decay[10,30–33]. The absolute QY was determined to be 66, 72, 62, and 54% in ethanol for the high color-purity B-, G-, Y-, and R-NBE-T-CQDs with the corresponding excitation wavelength being 460, 500, 520, and 580 nm, respectively. Notably, these QYs are among the highest values for CQDs reported to date, despite the fact that almost no fluorescence was detected in the macroscopic solid powder state due to the strong π–π interactions between the highly crystalline NBE-T-CQDs with a large π-conjugated structure. What is more, the NBE-T-CQDs also show the rarely seen strong high color-purity two-photon fluorescence (TPF) from blue to red (FWHM of 29 nm) (Supplementary Figs. 8-14), which may enable them to be used as excellent optical-gain media for high-performance frequency-up-conversion tunable lasers[38].

Bandgap energies of the NBE-T-CQDs were further calculated using the equation $E_g^{opt} = 1240/\lambda_{edge}$, where $\lambda_{edge}$ is the onset value of the first excitonic absorption peaks in the direction of longer wavelengths. The calculated bandgap energies gradually decreased from 2.63 to 2.07 eV with the excitonic emission peak red-shifting from 472 to 598 nm and the size increasing from 1.9 to 3.9 nm (Supplementary Fig. 15), further demonstrating the obvious size-dependent property of bandgap energies[18]. Meanwhile, the up-shifted highest occupied molecular orbital (HOMO) levels from −5.18 to −4.92 eV determined by means of ultraviolet photoelectron spectroscopy (UPS) and the down-shifted lowest unoccupied molecular orbital (LUMO) levels from −2.55 to −2.85 eV (Supplementary Figs. 16-18, Supplementary Table 1) of the NBE-T-CQDs from blue to red directly reveal that the quantum confinement effect dominates the electronic and optical properties of the NBE-T-CQDs.

To further scrutinize the bright and high color-purity excitonic emission of the NBE-T-CQDs from the perspective of transfer and recombination dynamics of the photogenerated charges, femtosecond transient absorption (fs-TA) spectroscopy measurement was carried out at 400 nm excitation. The TA spectra of B-NBE-T-CQDs are depicted in Fig. 2b in pseudo-3D with probe in

the 430–710 nm range and scan delay time from 0.1 ps to 2 ns. The negative (blue) features from 430 to 550 nm correspond to the ground state bleaching (GSB) and stimulated emission (SE) according to the steady-state absorption and PL spectra, and the relatively weaker positive (red) features from 600 to 710 nm correspond to the excited state absorption (ESA). The TA spectra at different time delays are shown in Fig. 2c. The negative peaks of SE centered at 466 and 524 nm gradually increase in the first picosecond. The kinetic traces at different wavelengths as a function of delay time are also presented in Supplementary Fig. 19. To unravel the detailed relaxation channels of the excited carriers of B-NBE-T-CQDs, global analyses were performed on the TA data, and four distinctive decay components were derived. The four fitted lifetimes of carriers are 0.54 ± 0.01 ps, 31.5 ± 0.8 ps, 77 ± 2 ps, and 7.3 ± 0.08 ns, respectively. The fitted decay associated difference spectra (DADS) are shown in Fig. 2d. It can be observed that during the first lifetime, the GSB decays accompanied by the rise of SE, then the signal of SE continues to increase within the second lifetime, but in the third and fourth lifetime, the SE signal decays. The weaker DADS at around ESA were enlarged to reveal more clearly the changes of different components. In order to analyze the four different carrier relaxation channels, the DADS at different wavelengths are normalized to evaluate the proportion of the decay dynamics (Supplementary Fig. 20). Different regions with distinct relaxation dynamics can be clearly observed as follows: the percentage of both the first and second component is zero at about 468 nm; the percentage of the second component at about 506 and 605 nm is zero; the percentage of the fourth component at about 598 nm is zero; the percentage of the third component at about 590 nm and from 632 to 700 nm are zero. On the basis of the different regions and DADS, we can ascribe the four components to the corresponding relaxation channels. Since the pump (400 nm) is higher in energy than the bandgap of B-NBE-T-CQDs, the excited carriers in the sp² cluster have excess energy after excitation, and will experience Coulomb-induced thermalization within the first few tens of femtoseconds, which is shorter than our instrumental response time (about 100 fs)[39]. The hot carriers will release the excess energy into the surrounding environment via optical phonon scattering (0.54 ps)[40] and acoustic phonon

scattering (31.5 ps)[41]. Part of the cooled carriers, whose dyanmics is distributed at about 598 nm, will experience nonradiative transition into the ground state within 77 ps. The remaining part will emit fluorescence (7.3 ns) via recombination of electrons and holes, and the ESA of this part is mainly distributed at 590 nm and 632–700 nm. Intriguingly, the strong emission of the NBE-T-CQDs is directly demonstrated here by the much higher amplitude of emissive component than that of nonradiative decay component, with the latter accounting for only a small percentage of about 15–20% (Fig. 2d, Supplementary Fig. 20). Furthermore, contrary to the complex nonradiative excited-state relaxation processes commonly responsible for the broadening of PL peaks, the quite simple excited state relaxation channels we obtained from the TA spectra naturally explain the high color-purity excitonic emission of NBE-T-CQDs[42,43].

To acquire more intrinsic characteristics of the photogenerated excitons in the high color-purity NBE-T-CQDs, temperature-dependent PL spectra were also recorded, and the resulting emission narrowing was analyzed, as has long been used to assess mechanisms of electron–phonon coupling in a wide range of bandgap emitting inorganic QDs[44,45]. As the temperature is decreased from 295 to 85 K, all the PL peaks of the NBE-T-CQDs show continuous narrowing and blue-shift (Fig. 2e, Supplementary Figs. 21-22). Remarkably, the PL spectra acquired at 85 K exhibit extremely narrow FWHM of 16, 11, 16, and 9 nm for the B-, G-, Y-, and R-NBE-T-CQDs, respectively (Fig. 2f), indicating that the narrow emission is attributable to the lowest state free-excitonic emission with negligible trap states. As shown in Fig. 2g, the FWHM reduces from 164.7 (30 nm) to 95.4 meV (16 nm) and the emission peak energy of the B-NBE-T-CQDs shifts toward higher energy from 2.627 to 2.725 eV with decreasing temperature from 295 to 85 K. The narrowed FWHM and blue-shifted emission peaks of the NBE-T-CQDs with decreasing temperature are well described by traditional empirical Varshni models, which can be explained by the reduced electron–phonon coupling due to the restricted structural vibrations and distortions at the lower temperatures[44–46]. It has been demonstrated that the electron–phonon coupling resulting from structural vibrations and distortions plays a dominating role in determining the FWHM of the PL spectra of inorganic QDs[44,45]. Therefore, it is reasonable to conclude that the dramatically reduced electron–phonon coupling demonstrated by the temperature-dependent PL spectra leads to the high color-purity, free-excitonic emission of the NBE-T-CQDs.

Apart from the temperature-dependent PL peak wavelengths and FWHM, the integrated PL intensity of NBE-T-CQDs shows a slight decrease with increasing temperature (Fig. 2h), which can be ascribed to thermally activated exciton dissociation and nonradiative trapping[47]. Importantly, the thermal quenching of the integrated PL intensity of the NBE-T-CQDs with increasing temperature from 175 to 295 K is <20%, indicating the high thermostability and the minimal nonradiative recombination centers or defects. To extract the important physical parameter of exciton binding energy, we plot in Fig. 2h the integrated PL emission intensity as a function of temperature (175–295 K). The curves can be fitted using the following equation:

$$I(T) = \frac{I_0}{1 + Ae^{-E_b/k_B T}}$$

where $I_0$ is the intensity at 0 K, $E_b$ is the exciton binding energy, and $k_B$ is the Boltzmann constant. From the fitting analysis, the NBE-T-CQDs have a relatively large exciton binding energy of 139.2, 128.2, 110.8, and 100.6 meV for the B-, G-, Y-, and R-NBE-T-CQDs, respectively (Supplementary Figs. 23-26), which is even larger than those of many inorganic QDs and thus contributes to

the high color-purity of the NBE-T-CQDs. The relatively large exciton binding energy of NBE-T-CQDs decreased from 139.2 (B-NBE-T-CQDs) to 100.6 meV (R-NBE-T-CQDs) in a way conforming to the corresponding bandgap decrease, which is probably due to confinement effect associated with the unique quantum-sized high crystalline triangular graphene structure involving the size-dependent coulomb interaction[48] and bandgap energy[49]. To the best of our knowledge, this is the first time that the important physical parameters of exciton binding energy are obtained for the CQDs. Moreover, besides the high thermo-stability concluded from the temperature-dependent PL spectra, the NBE-T-CQDs also showed more robust photostability than the best protected core-shell inorganic QDs such as CdZnS@ZnS and organic dyes such as fluorescein (2-(6-hydroxy-3-oxo-3h-xanthen-9-yl)-benzoicacid) under continuous radiation of a UV lamp for 10 h (Supplementary Fig. 27), giving them more competitive edges for LED applications.

**Structural characterizations.** Detailed structural characterizations were conducted so as to further shed light on the high color-purity and reveal its intrinsic relation with the structure of the NBE-T-CQDs. It should be emphasized again that the aberration-corrected HAADF-STEM images of the NBE-T-CQDs were obtained for the first time, which clearly demonstrate the high crystalline triangular structure of the NBE-T-CQDs (Fig. 3a). The wide-area TEM images of NBE-T-CQDs all show a narrow size distribution with the distinctive highly crystalline triangular structure as highlighted by the white contour lines (Fig. 3b, Supplementary Fig. 28). The sixfold symmetric fast Fourier transform (FFT) patterns of the HRTEM images as well as the identical well-resolved lattice fringes with a spacing of 0.21 nm corresponding to the (100) inter-planar spacing further demonstrate the almost defect-free graphene crystalline structure of the NBE-T-CQDs (Supplementary Fig. 29)[3,7,10,18]. The gradually increased average sizes from 1.9 to 3.9 nm are well consistent with the red-shifted emission colors from blue to red of the triangular CQDs, manifesting the quantum confinement effect[18]. The X-ray powder diffraction (XRD) patterns of the NBE-T-CQDs show a narrow (002) peak centered at around 24°(Fig. 3c) in contrast to the ultrabroad (002) peak of the previously reported CQDs[9–11,15–18], which confirms the graphene structure of the NBE-T-CQDs with high crystallinity. The high degree of gra-phitization of the NBE-T-CQDs is reflected in their Raman spectra (Supplementary Fig. 30), where the crystalline G band at 1615 cm$^{-1}$ is much stronger than the disordered D band at 1380 cm$^{-1}$ with a large G to D intensity ratio ($I_G/I_D$) of about 1.5–1.8, indicating the high quality of the graphene structure of the NBE-T-CQDs, and is well consistent with the high crystalline graphene structure determined by HAADF-STEM and HRTEM images. To the best of our knowledge, the $I_G/I_D$ values of the NBE-T-CQDs are among the largest ever reported to date for CQDs[18].

In the $^1$H-nuclear magnetic resonance (NMR) spectra (acetone-d6, ppm) (Fig. 3d, Supplementary Figs. 31-33), apart from the obvious aromatic hydrogen signals detected in the range of 7–8 ppm, active hydrogen signals from hydroxy groups with broad peaks as black arrow indicated in Fig. 3d are also observed[50]. Moreover, $^{13}$C-NMR spectra (methanol-d4, ppm) of the NBE-T-CQDs (Fig. 3e, Supplementary Figs. 34-37) further confirm the functionalization with pure electron-donating hydroxy groups at the edge sites. The clearly observed resonance signals in the range of 155 to 170 ppm are indicative of the sp$^2$ carbon atoms bonded with hydroxy groups at the edge sites of the NBE-T-CQDs[50]. In addition, the emerging numerous signals observed in the range of 115–140 ppm in the $^{13}$C-NMR spectra compared with that of PG (Supplementary Fig. 34) further

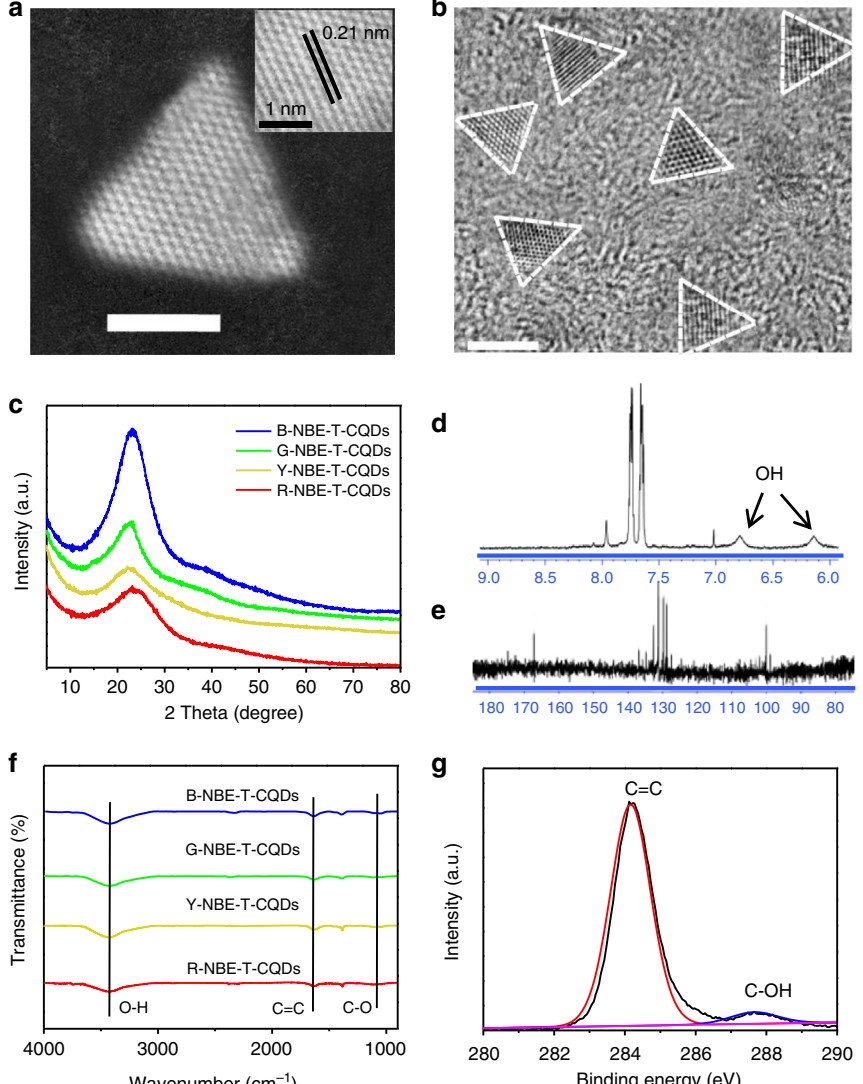

**Fig. 3** Structural characterizations of the NBE-T-CQDs. **a** The typical aberration-corrected HAADF-STEM image of R-NBE-T-CQDs (the inset is the corresponding high-resolution image). **b** The wide-area TEM image of G-NBE-T-CQDs. Scale bar, 2 nm. (The triangular projections are highlighted by white contour lines). XRD patterns (**c**) and FT-IR spectra (**f**) of NBE-T-CQDs. $^1$H-NMR (**d**), $^{13}$C-NMR (**e**), and C1s (**g**) spectra of B-NBE-T-CQDs

demonstrates the formation of intact sp$^2$ domains during the synthesis of the NBE-T-CQDs. Note that the NBE-T-CQDs with different emission colors all exhibit similar Fourier transform infrared (FT-IR) spectra, attesting to their similar chemical compositions. Besides, the strong stretching vibration bands of O-H, C=C, and C-O, characteristic of the NBE-T-CQDs are also observed at 3435, 1630, and 1100 cm$^{-1}$, respectively (Fig. 3f, Supplementary Fig. 38). The X-ray photoelectron spectroscopy (XPS) surveys further confirm the FT-IR data and demonstrate that the NBE-T-CQDs all have the same elemental composition (i.e. C and O) (Supplementary Fig. 39). The deconvoluted high-resolution XPS spectra for C1s (Fig. 3g, Supplementary Fig. 40) and O1s (Supplementary Fig. 41) indicate that they contain the same C=C and O-H chemical bonds. The similar structure and chemical compositions indicate that the optical properties of the NBE-T-CQDs are dominated by their sizes due to the quantum confinement effect[18].

Taken together, it is evident that under the given reaction conditions in tandem with the elaborate separation and purification, the as-prepared NBE-T-CQDs are highly crystalline and have a unique triangular structure functionalized with pure

electron-donating hydroxyl groups at the edge sites. Significantly, the NBE-T-CQDs show almost no surface defects due to the highly crystalline structure and the absence of such electron-withdrawing oxygen-containing groups as carboxyl, carbonyl and epoxy groups, which could act as surface defects and trap sites usually observed in conventional CQDs. The sharply reduced electron–phonon coupling confirmed by detailed optical characterizations coupled with the absence of surface defects due to the unique triangular structure of the NBE-T-CQDs yield the strong high color-purity excitonic emission[27–33].

**Theoretical investigation.** Our proposal that the unique highly crystalline triangular structure functionalized with pure electron-donating hydroxyl groups at the edge sites are responsible for the high color-purity excitonic emission of the NBE-T-CQDs was further confirmed by DFT theory calculations. The optical properties of different kinds of model CQDs with triangular structure consisting of 4, 10, and 19 fused benzene rings functionalized with electron-donating hydroxyl groups (T-CQDs-OH) (Fig. 4a, e, i) or electron-withdrawing carboxyl groups

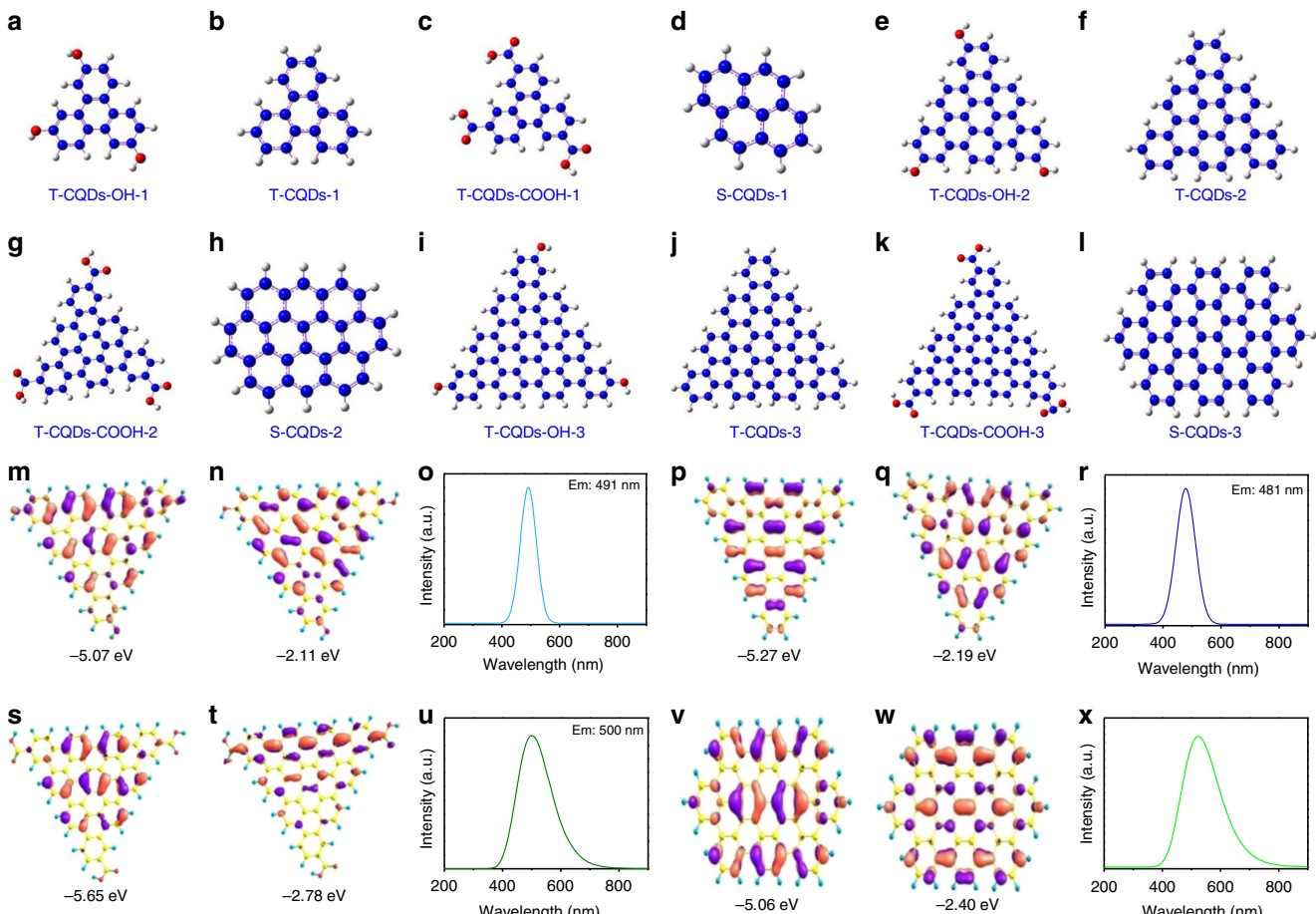

**Fig. 4** Time-dependent DFT calculation results. The triangular structure model CQDs consisting of 4, 10, and 19 fused benzene rings: (1) functionalized with pure electron-donating hydroxyl groups (T-CQDs-OH-1 (**a**), T-CQDs-OH-2 (**e**), T-CQDs-OH-3 (**i**)), (2) without functionalization (T-CQDs-1 (**b**), T-CQDs-2 (**f**), T-CQDs-3 (**j**)), (3) functionalized with pure electron-withdrawing carboxyl groups (T-CQDs-COOH-1 (**c**), T-CQDs-COOH-2 (**g**), T-CQDs-COOH-3 (**k**)). The square-like structure model CQDs without functionalization consisting of 4 (S-CQDs-1 (**d**)), 10 (S-CQDs-2 (**h**)) and 20 (S-CQDs-3 (**l**)) fused benzene rings. The calculated HOMO (**m**, **p**, **s**, **v**), LUMO (**n**, **q**, **t**, **w**), and PL spectra (**o**, **r**, **u**, **x**) of T-CQDs-OH-3, T-CQDs-3, T-CQDs-COOH-3, and S-CQDs-3, respectively

(T-CQDs-COOH) (Fig. 4c, g, k) or without functionalization (T-CQDs) (Fig. 4b, f, j) as well as the square-like structure consisting of 4, 10, and 20 fused benzene rings (S-CQDs) (Fig. 4d, h, l) were all calculated for comparison (Supplementary Fig. 42-45, Supplementary Table 2-14)[51]. Remarkably, the T-CQDs-OH and T-CQDs show distinct charge delocalization and optical properties compared with T-CQDs-COOH and S-CQDs. The degree of delocalization of the HOMO and LUMO can be qualitatively judged, for instance, by their corresponding electron cloud density distributions around the whole molecular structure. In this sense, a more uniformly distributed electron cloud density would indicate a higher degree of delocalization. Clearly, the calculated electron cloud densities of HOMO and LUMO for both T-CQDs-OH and T-CQDs are more uniformly distributed across the whole molecular structure than those for T-CQDs-COOH and S-CQDs (Fig. 4m–x, Supplementary Fig. 46-52). Therefore, it can be reasonably concluded that the HOMOs and LUMOs of T-CQDs-OH and T-CQDs show higher degrees of delocalization than those of T-CQDs-COOH and S-CQDs. For the optical properties of different kinds of model CQDs, apart from the differences in the emission peaks and energy levels such as the HOMO, LUMO, and bandgap energies (Fig. 4m–x, Supplementary Fig. 46-52), the FWHM of the PL spectra of T-CQDs-OH is slightly smaller than that of T-CQDs, but much smaller than those of T-CQDs-COOH and S-CQDs. At the fundamental level, it stands as a universal

law that the T-CQDs-OH and T-CQDs show much higher color-purity emission than T-CQDs-COOH and S-CQDs, which is observed by all these different kinds of model CQDs consisting of different number of fused benzene rings (Supplementary Fig. 53-56). Take for example, the PL spectra of T-CQDs-OH-3 consisting of 19 fused benzene rings shows a narrow FWHM of 64 nm (Fig. 4o), which is slightly smaller than that of T-CQDs-3 (FWHM: 73 nm) (Fig. 4r), but is even smaller than half of the FWHM of the PL spectra of T-CQDs-COOH-3 (FWHM: 135 nm) (Fig. 4u) and S-CQDs-3 (FWHM: 149 nm) (Fig. 4x). More significantly, the increased thermodynamic stability of T-CQDs-OH compared with T-CQDs and S-CQDs are also demonstrated by theoretical calculations (Supplementary Table 15), which directly show the outstanding structural stability of T-CQDs-OH.

These elaborately designed theoretical calculations demonstrate that the triangular structure and electron-donating hydroxyl groups play significant roles in determining the high color-purity of the NBE-T-CQDs which can be explained in detail as follows: first, the higher degree of delocalization leads to higher structural stability of the unique triangular structure of the NBE-T-CQDs, which in turn results in dramatically reduced electron–phonon coupling. This contributes to the high color-purity excitonic emission and narrow FWHM of PL spectra of NBE-T-CQDs as demonstrated by the temperature-dependent PL spectra. Second, the pure electron-donating hydroxyl groups at the edge sites of

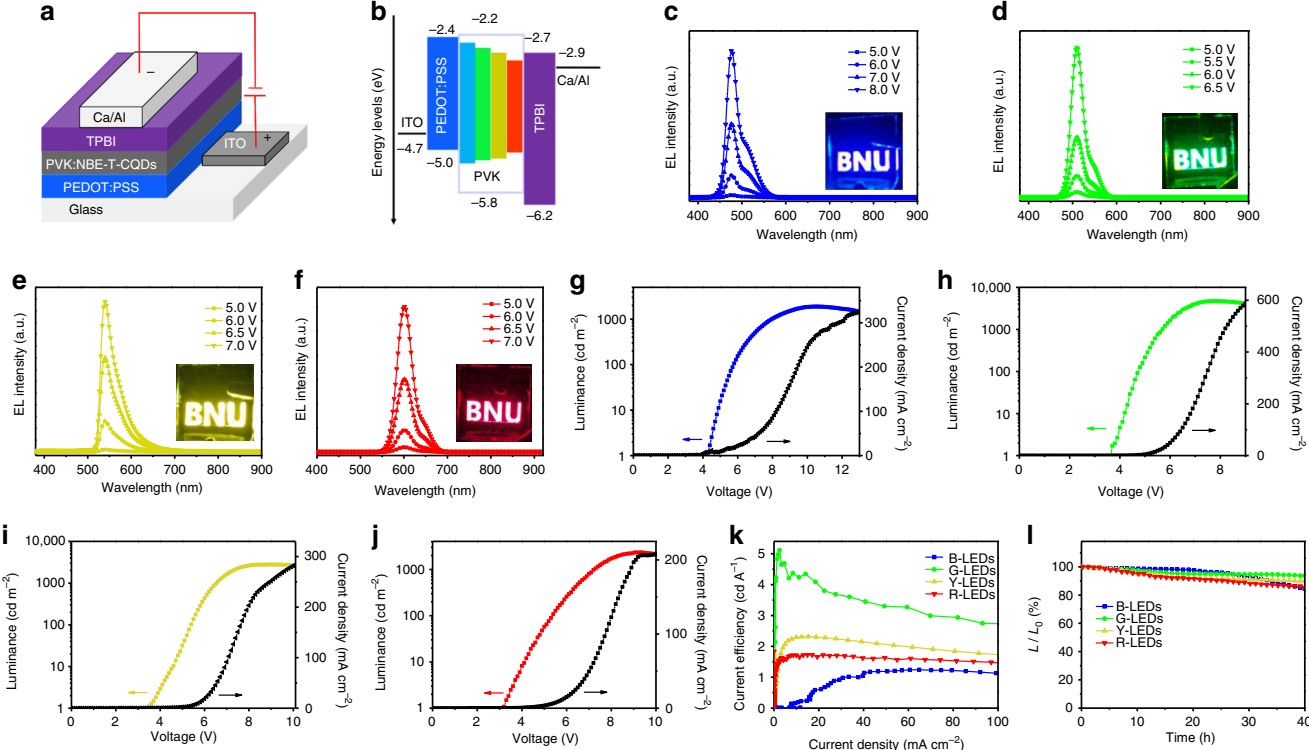

**Fig. 5** LED structure, energy diagram, and performance characterization. The device structure (**a**) and energy level diagram (**b**) of the NBE-T-CQDs-based LEDs. EL spectra of the B- (**c**), G- (**d**), Y- (**e**), and R-LEDs (**f**) at different bias voltage, respectively. (Insets are the operation photographs of the B-, G-, Y-, and R-LEDs with the logo of BNU). The maximum luminance–current–voltage (L–I–V) characteristic of B- (**g**), G- (**h**), Y- (**i**), and R-LEDs (**j**), respectively. The current efficiency versus current density (**k**) and the stability plots (**l**) of the B-, G-, Y-, and R-LEDs

the NBE-T-CQDs can also greatly increase the π electron cloud density and facilitate the pure radiative recombination of confined electrons and holes. On the contrary, the electron-withdrawing carboxyl groups on $sp^2$-hybridized carbons can induce significant local distortions as well as acting as surface defects simultaneously, which could trap carriers and finally result in dramatically increased FWHM of the PL spectra of CQDs (Fig. 4s–u, Supplementary Fig. 54, 56). Summarizing the above, the theoretical investigation demonstrates that the unique highly crystalline triangular structure functionalized with pure electron-donating hydroxyl groups at the edge sites show highly delocalized charges, outstanding structural stability, and thus dramatically reduced electron–phonon coupling, which are responsible for the high color-purity excitonic emission of the NBE-T-CQDs.

**LED performance**. The bright and high color-purity excitonic emission of the NBE-T-CQDs has prompted us to exploit their applications in LEDs for the development of next-generation display technology. A conventional simple structure was used for fabrication of the LEDs from blue to red with the NBE-T-CQDs blended poly(N-vinyl carbazole) (PVK) as the active emission layer, as shown in Fig. 5a. PVK was selected as host material due to its excellent hole transporting properties as well as favorable film forming properties[52]. Atomic force microscopy (AFM) measurements show that the NBE-T-CQDs blended PVK film has a smooth and uniform surface coverage with small root-mean-square (rms) roughness in the range of 1.31–1.80 nm (Supplementary Fig. 57). The QY of B-, G-, Y-, and R-NBE-T-CQDs blended PVK films were determined to be about 56, 62, 48, and 42%, respectively. The device structure consists of, from the bottom up, a ITO glass substrate anode, a poly(3,4-ethylene-dioxythiophene): poly(styrene-sulfonate) (PEDOT:PSS) hole

injection layer (HIL), an active NBE-T-CQDs:PVK blended emission layer, a 1,3,5-tris(N-phenylbenzimidazol-2-yl) benzene (TPBI) electron transport layer (ETL), and a Ca/Al double-layered cathode. The thickness of PEDOT:PSS, PVK:NBE-T-CQDs, and TPBI layers in LED devices are determined to be about 24 to 28, 19 to 22, and 30 to 32 nm, respectively, as confirmed by the cross-sectional TEM images and the EDX maps of LED devices (Supplementary Fig. 58-59). As observed in the energy level diagram of the NBE-T-CQDs-based LEDs shown in Fig. 5b, the HOMO and LUMO energy levels of NBE-T-CQDs are located within those of the PVK, and have a small energy barrier for charge injection from both electrodes to the PVK host[52]. Then, the electrons and holes can be efficiently transferred from PVK to NBE-T-CQDs emitter in the active layer. The transferred electrons and holes can undergo radiative recombination in the NBE-T-CQDs, giving rise to the electroluminescence (EL) [52].

The EL spectra of the NBE-T-CQDs-based LEDs are presented in Fig. 5c–f. They exhibit peak wavelengths at 476, 510, 540, and 602 nm, respectively, and are in good agreement with the PL emission peaks measured in the solution, indicating the excellent dispersion of the NBE-T-CQDs in the host material of PVK (Supplementary Fig. 60-61). More significantly, the NBE-T-CQDs-based LEDs show high color-purity EL emission with narrow FWHM of 30, 32, 38, and 39 nm for the B-, G-, Y-, and R-LEDs, respectively, which is even comparable to the well-developed high color-purity inorganic QDs-based LEDs[53,54]. The operation photographs with Beijing Normal University (BNU) logo (insets of Fig. 5c–f, Supplementary Fig. 61) display the close-up view of the bright, uniform and defect-free surface high color-purity EL emission from blue, green, yellow, to red of the NBE-T-CQDs-based LEDs. The apparently voltage-independent emission color (Fig. 5c–f) indicates the high color-

**Table 1 The performances of high color-purity NBE-T-CQDs-based LEDs from blue to red**

| LEDs | PL/FWHM (nm) | EL/FWHM (nm) | $V_{ON}$ (V) | $L_{max}$ (cd m$^{-2}$) | $\eta_c$ (cd A$^{-1}$) |
|---|---|---|---|---|---|
| B-LEDs | 472/30 | 476/30 | 4.3 | 1882 | 1.22 |
| G-LEDs | 507/29 | 510/32 | 3.7 | 4762 | 5.11 |
| Y-LEDs | 538/30 | 540/38 | 3.5 | 2784 | 2.31 |
| R-LEDs | 598/30 | 602/39 | 3.1 | 2344 | 1.73 |

stability of the LEDs, which is of great significance for display technology. To the best of our knowledge, this is the first report for the fabrication of high color-purity CQDs-based LEDs (FWHM <40 nm) with stable emission color from blue to red with the NBE-T-CQDs blended PVK as active emission layer.

The typical luminance and current density curves as a function of applied voltage for the NBE-T-CQDs-based LEDs are shown in Fig. 5g-j and Supplementary Fig. 62. The performances of the LEDs are summarized in Table 1. The $V_{on}$, defined as the bias voltage applied to a LEDs producing a brightness of 1 cd m$^{-2}$, decreased from 4.3 to 3.1 V for the LEDs from blue to red, which are much lower than the previous reported CQDs-based LEDs[18,52] due to the maching energy levels of the related materials (Fig. 5b). The $L_{max}$ and $\eta_c$ reach about 4762 cd m$^{-2}$ and 5.1 cd A$^{-1}$ for green LEDs (G-LEDs) (Fig. 5h, k), respectively, which are the best performance ever reported for the CQDs-based LEDs, and is about 50 and 110 times higher than our previous reported G-LEDs which directly used bandgap fluorescent CQDs as the active emission layer without using the PVK host[18]. Other colored high color-purity LEDs based on the NBE-T-CQDs fabricated with the same device structure also show high-performance with $L_{max}$ reaching 1882, 2784, and 2344 cd m$^{-2}$ for the B-, Y-, and R-LEDs coupled with corresponding $\eta_c$ of 1.22, 2.31, and 1.73 cd A$^{-1}$ (Fig. 5g, i–k, Supplementary Fig. 63), respectively, which are to some extent comparable to the QDs-based LEDs (Supplementary Table 16). Compared with the NBE-T-CQDs-based LEDs fabricated without using the PVK polymer host, the one with PVK exhibited greatly improved $L_{max}$ and $\eta_c$ by 12 to 25 and 17 to 28 times, respectively, as shown in Supplementary Table 17. Besides the bright fluorescence inherent to the NBE-T-CQDs, the hole-transport PVK polymer as a host material also contributed to the remarkable performance of our LEDs due to the resulting optimized charge balance in the emission layer. As an ultrastable feature of the fluorescence of the NBE-T-CQDs, the high color-purity NBE-T-CQDs-based LEDs exhibit outstanding ambient stability. After operation for 40 h, more than 85% of initial luminance ($L_0$: 500 cd m$^{-2}$) are retained (Fig. 5l) without degradation of the high color-purity (Supplementary Fig. 64). Moreover, the LEDs also show high stability at extremely high voltage, further demonstrating the great potential applications of the NBE-T-CQDs-based LEDs for the development of next-generation display technology (Supplementary Fig. 65).

## Discussion

We report the subversive demonstration of high color-purity NBE-T-CQDs (FWHM of 29–30 nm) from blue to red with a QY up to 54–72%. The NBE-T-CQDs were prepared by facilely controlling the fusion and carbonization of three-fold symmetric PG triangulogen which possesses a unique structure with three highly reactive hydrogen atoms at the three meta-positions activated by three electron-donating hydroxyl groups in a single molecule. Detailed structural and optical characterizations together with elaborate theoretical calculations revealed that the

unique rigid triangular structure, molecular purity, crystalline perfection and most importantly, the resulting weak electron–phonon interaction of the NBE-T-CQDs surrounded by hydroxy groups are the key points to the high color-purity. The multicolored LEDs based on the NBE-T-CQDs demonstrated high color-purity (FWHM of 30–39 nm), a $L_{max}$ of 1882–4762 cd m$^{-2}$ and $\eta_c$ of 1.22–5.11 cd A$^{-1}$, rivaling the well-developed inorganic QDs-based LEDs. Moreover, the LEDs demonstrate outstanding stability. We anticipate that this work will inspire further research on and more optimizations of the charge injection (holes and electrons) as well as better designed devices, leading to a greatly improved performance for the high color-purity NBE-T-CQDs-based LEDs ideal for next-generation display technology. Detailed follow-up work along this line is underway in our laboratory.

## Methods

**Synthesis of high color-purity NBE-T-CQDs**. The highly tunable and high color-purity NBE-T-CQDs from blue to red can be synthesized by solvothermal treatment or refluxing of phloroglucinol (PG) in various common solvents. In a typical preparation procedure for the synthesis of blue and green NBE-T-CQDs: PG (500 mg) was dissolved in ethanol (10 mL). The clear precursor solution after 10 min ultrasonic dissolving was then transferred to a poly(tetrafluoroethylene) (Teflon)-lined autoclave (25 mL) and heated at 200 °C for 9 and 24 h. After the reaction, the reactors were cooled to room temperature by water or naturally. For yellow and red NBE-T-CQDs: PG (500 mg) was dissolved in ethanol (10 mL), followed by adding concentrated hydrochloric acid or sulfuric acid (2 mL) as catalyst. The clear precursor solution was then transferred to a poly(tetrafluoroethylene) (Teflon)-lined autoclave (25 mL) and heated at 200 °C for 2 and 5 h. After the reaction, the reactors were cooled to room temperature by water or naturally, the hydrochloric acid was removed by direct heating of the solution for 30 min, or the solution was neutralized by sodium hydroxide and the supernatant was collected by centrifugation. The solvent ethanol for the synthesis of NBE-T-CQDs can also be changed to various other common solvents such as formamide, N, N-dimethyl formamide, water, and so on. In addition, refluxing of PG in various common solvents such as ethanol, formamide, N, N-dimethyl formamide can also produce tunable fluorescent NBE-T-CQDs through optimization of reaction conditions. For example, the red NBE-T-CQDs can be prepared by refluxing of PG in ethanol with concentrated hydrochloric acid or sulfuric acid as catalyst. The green and yellow NBE-T-CQDs can be prepared by refluxing of PG in N, N-dimethyl formamide and formamide with small amount of concentrated hydrochloric acid as catalyst. The NBE-T-CQDs are purified via silica column chromatography using a mixture of dichloromethane and methanol as the eluent. During the silica column chromatography purification process, the polarity of eluent should be changed dynamically by changing the volume ratio of dichloromethane to methanol in order to ensure the effectiveness of the separation and purification process. Specifically, the volume ratio was dynamically changed during the silica column chromatography purification process from 6:1 to 2:1 (blue NBE-T-CQDs), 10:1 to 4:1 (green NBE-T-CQDs), 16:1 to 8:1 (green NBE-T-CQDs), and 25:1 to 10:1 (red NBE-T-CQDs). Typically, the silica column chromatography purification process should be repeated several times in order to obtain pure NBE-T-CQDs.

**Quantum yield measurements**. An absolute method, using Varian FLR025 spectrometer equipped with a 120 mm integrating sphere, was employed to determine the QY of NBE-T-CQDs. We conducted the test light from FLR025 spectrometer to the sphere. The QY was determined by the ratio between photons emitted and absorbed by NBE-T-CQDs. The ethanol solution was placed in a UV quarts cuvette with a light path of 10 mm to measure its QY, while the solvent ethanol filled in the quarts cuvette was used as a blank sample for the reference measurement. The spectral correction curve which relates to the sensitivity of the monochromator, detector, sphere coating, and optics to wavelength was provided by Edinburgh Instruments.

**Femtosecond transient absorption setup**. A regeneratively amplified Ti:sapphire laser system (Coherent Libra, 50 fs, 1 kHz) provides the fundamental light source. The pump pulse (400 nm) is generated by focusing a portion of fundamental light into BBO crystal. In order to avoid the influence of rotational relaxation effects on dynamics, the polarization of pump pulse is randomized by depolarizing plate. The other fundamental pulse provides broadband probe pulse (white light continuum) that is produced by focusing 800 nm fundamental light into sapphire plate (3 mm). The pump and probe beams are overlapped in the sample with crossing areas of 600 and 150 µm. After passing through the sample, the probe pulse is focused into optical fiber that is coupled to spectrometer (AvaSpec-1650F). The energy of 400 nm excitation pulse is adjusted to about 1 µJ per pulse by a neutral density optical filter. The pump pulse is chopped at 500 Hz to acquire pumped (signal) and un-pumped (reference) probe spectra, and the ΔOD spectrum can be obtained by

processing them. The solutions are placed in 2 mm optical path length quartz cuvette. Both the instrument response function (100 fs) and temporal chirp in the probe light are determined by measuring the cross modulation of solvent. The group velocity dispersion effect is corrected by home-made chirp program. For each measurement, the pump-probe delay scan is repeated three times to give the averaged experiment data.

**Ultraviolet photoelectron spectroscopy measurement**. UPS measurement was performed with an hv = 21.22 eV, He I source (AXIS ULTRA DLD, Kratos). The analysis room vacuum was $3.0 \times 10^{-8}$ Torr, and the bias voltage for measurement was −9 V. The NBE-T-CQDs thin films were prepared from spin-coating on ITO substrates for UPS measurement.

**Characterization method**. A JEOL JEM 2100 transmission electron microscope (TEM) was used to investigate the morphologies of the NBE-T-CQDs. X-ray diffraction (XRD) patterns were carried out by an X-ray diffraction using Cu-Ka radiation (XRD, PANalytical X'Pert Pro MPD). Absorption spectra were recorded on UV-2450 spectrophotometry. The fluorescence spectra of NBE-T-CQDs were measured on a PerkinElmer-LS55 luminescence spectrometer with slit width at 2.5–2.5 nm. The photographs were taken with camera (Nikon, D7200) under UV light excited at 365 nm (UV light: SPECTROLINE, ENF-280C/FBE, 8 W). The FT-IR spectra were measured using a Nicolet 380 spectrograph. X-ray photoelectron spectroscopy (XPS) was performed with an ESCALab220i-XL electron spectrometer from VG Scientific using 300 W Al Ka radiation. The Raman spectrum was measured using Laser Confocal Micro-Raman Spectroscopy (LabRAM Aramis). Low-temperature-dependent PL spectra measurements were performed in the temperature range of 85–295 K using a liquid nitrogen cooler. The TPF spectra of NBE-T-CQDs ethanol solution placed in the 1 cm fluorescence cuvette were recorded on the fiber spectrometer (Ocean Optics USB2000 CCD) with a Ti:sapphire femtosecond laser (Spitfire, Spectra-Physics, 100 fs, 80 MHz, 880 nm) for excitation.

**STEM-HAADF images characterization**. A JEM-ARM200F transmission electron microscope (TEM) was used to investigate the STEM-HAADF images of the NBE-T-CQDs. Ultrathin carbon film supported by a lacey on a 400 mesh copper grid (product no. 01824, bought from Beijing Xinxing Braim Technology Co., Ltd) was used to disperse the NBE-T-CQDs. The purified diluted NBE-T-CQDs ethanol solution with 5 μL was dropped on the surface of ultrathin carbon film, and then dried at room temperature. Finally, the STEM-HAADF images of NBE-T-CQDs samples were measured at 200 KV.

**Theoretical calculations**. All the optical properties of different kinds of model CQDs were calculated using time-dependent density functional theory (TDDFT) method as implemented in the Gaussian09 software package. The 6-311G(d) basis set was selected to combine with the functional B3LYP throughout all calculations (B3LYP/6-311G(d)). The first excited state was optimized in vacuum to calculate the emission energy (wavelength) which is the energy difference between the ground and the first excited state. The pink and violet colors in the HOMO and LUMO molecular orbitals represent the positive and negative phases of the molecular orbital wavefunctions.

**Device fabrication and characterization**. Indium-tin-oxide (ITO)-coated glass substrates was cleaned ultrasonically in organic solvents (acetone and isopropyl alcohol), rinsed in deionized water, and then dried in an oven at 150 °C for 10 min. The substrates were cleaned with a UV-ozone treatment to enrich the ITO surface with oxygen to increase the ITO work function. The poly(3,4-ethylenedioxythiophene): poly(styrenesulfonate) (PEDOT:PSS) hole injection layer (HIL) was spin-coated at 2000 rpm for 35 s on the ITO with a thickness of about 30 nm, followed by annealing in an oven at 150 °C for 15 min. Subsequently, the emissive layer of NBE-T-CQDs blended poly(N-vinyl carbazole) (PVK) was spin-coated at 3000 rpm for 45 s over the surface of PEDOT:PSS film from the mixed solution of o-dichlorobenzene and ethanol solution, followed by baking on a hot plate at 80 °C for 30 min to form the active region of the NBE-T-CQDs-based monochrome LEDs. Finally, the substrates were transferred to a vacuum chamber and a 30 nm thick 1,3,5-tris(N-phenylbenzimidazol-2-yl) benzene (TPBI) electron transport layer (ETL) was thermally deposited with base pressure of $3 \times 10^{-4}$ Pa. After that, a 20 nm Ca and 100 nm thick Al cathode was deposited using a shadow mask with 2 mm width. The active area of the devices was thus 4 mm². The thermal deposition rates for TPBI and Ca/Al are 1, 1, and 3 Å s$^{-1}$, respectively. PEDOT:PSS was used as a buffer layer on the anode mainly to increase the anode work function from 4.7 (ITO) to 5.0 eV and to reduce the surface roughness of the anode to obtain stable and pinhole-free electrical conduction across the device. TPBI was chosen as the ETL because of its good electron transport capability and its interfacial phase compatibility with the active emission layer. The thickness of films was measured using a Dektak XT (Bruker) surface profilometer and a spectroscopic ellipsometer (Suntech). The luminance–current–voltage (L–I–V) characteristics were measured using a computer-controlled Keithley 236 SMU and Keithley 200 multimeter coupled with a calibrated Si photodiode. Electroluminescence (EL) spectra were measured by an Ocean Optics 2000 spectrometer, which couples a linear charge-coupled device (CCD)-array detector ranging from 350 to 1100 nm.

**Data availability**. The data that support the findings of this study are available from the corresponding author upon reasonable request

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

## Acknowledgements

This work is supported by NSFC of China (21573019), the Major Research Plan of NSFC (21233003), Shenzhen Peacock Plan Program (KQTD2016053015544057), NSFC of China (11674124), National Basic Research Program of China (973 Program, Grant No. 2013CB922200), and the Fundamental Research Funds for the Central Universities.

## Author contributions

L.F. and F.Y. had the idea for and designed the experiments. L.F., Z.T., and S.Y. supervised the work. F.Y. conducted the synthesis and characterization of NBE-T-CQDs. Z.X. conducted the NMR characterization of NBE-T-CQDs. T.Y. participated in the purification of NBE-T-CQDs. T.Y. performed the theoretical calculations. L.S. measured the femtosecond transient absorption spectroscopy. L.S., F.Y., M.J., and A.C. analyzed the femtosecond transient absorption date. F.Y. carried out the LED device fabrication and characterizations. Z.T., and Z.W. participated in the LED device structure designation, fabrication, and characterizations. F.Y. wrote the first draft of the manuscript. L.F., Z.T. and S.Y. participated in data analysis and provided major revisions. All authors discussed the results and contributed to the manuscript.
