## [Peer Review File · Nature Communications]

Reviewers' Comments:

Reviewer #1:

Remarks to the Author:

In this manuscript (NCOMMS-18-00184), the authors suggested a very interesting synthetic idea and route to form stable crystalline structure and high quality triangular shaped CQDs which show both high color-purity with smaller FWHM in PL and higher thermostability than ever reported. It was also revealed that T-CQDs with electron-donor function group –OH are most efficient compared to those with electron accepting groups like COOH, C-O etc by reducing electron-phonon scattering through high delocalization. T-CQDs-based LED show highly bright luminance and current efficiency when PVK polymer was incorporated as host material.

This paper will be appropriate for publication in NCOMMs, but the following questions should be further clarified before acceptance.

1. What's the exact name of fluorescein used as a comparative reference in S26?
2. Exciton binding energy of ca. 139 meV for B-NBE-T-CQD decreased to that of ca. 100 meV of R-NBE-T-CQDs. Could explain why the exciton binding energies of T-CQDs were quite larger than any other inorganic (e.g. 60 meV for ZnO) and were reduced as the emission center is changed from blue to red?
3. In Fig. 4, how can the degrees of delocalization be defined in HOMO and LUMO levels for T-CQDs-OH, T-CQDs, T-CQDs-COOH, and S-CQDs by DFT calculation? And therefore how can they connect with the FWHM of PL spectra? What does the pink and violet colors of each (a), (b), (d) and (e) in S51-55 mean? Please add the explanation in figure caption or in text.
4. As the ref. [18], it will be also important to estimate the luminance, current efficiency of the pure (B, G, Y, G)-T-CQDs-OH based LED without being mixed with PVK polymer host. Then it should be compared how much it was improved when PVK host was used.

Reviewer #2:

Remarks to the Author:

The exploitation of new method for engineering high quality carbon quantum dots (CQDs) with multicolor and narrow bandwidth emission is very challenging and desired. In this manuscript, the authors reported preparation methods to achieve unprecedented triangular CQDs by a facile solvent thermal process. Meanwhile, the as-prepared CQDs exhibit tunable multicolor fluorescence emission with very narrow full-width-at-half-maximum (FWHM) (~30 nm) and high absolute quantum yield (up to 72%). The unique triangular structures of CQDs were fully characterized by TEM, Raman, XRD, NMR and XPS, especially high-angle annular dark-field scanning transmission electron microscopy (HAADF-STEM). In addition, the formation mechanism for the achievement of the triangular CQDs was also discussed. Based on the unprecedented structure and unique optical properties of the as-prepared CQDs, multicolored LEDs with high color-purity and stability renders this manuscript to be acceptable by Nature Communications, if the following questions can be addressed.

1. The reproducibility of preparation has to be confirmed and provided! We had followed the procedures that the authors described; unfortunately, we cannot obtain the same results as that of the authors reported in this manuscript. I suggest the authors to provide third-party results to prove the reproducibility of their method.
2. The preparation yield for the materials should be provided.
3. The relatively low resolution TEM image should also be provided, which can be used to illustrate the general morphologies of the as-prepared materials. In addition, AFM is also better to measure.
4. The wavelength value should be added in the x-axis in Figure 1h and 1i; the FL excitation spectra of the prepared CQDs should be measured and provided.
5. In Figure 1f, is the fluorescence image really obtained under the same single UV excitation

wavelength (365 nm)? This seems does not reasonable, because no absorption of the G-, Y-, and R-CQDs at 365 nm!

6. Please indicate the emission and excitation wavelengths of the decay spectra in the time-resolved decay curve measurement in Figure 2a.

7. Based on the reaction conditions and supposed reaction mechanism, CQDs with longer emission wavelengths (e.g., pure red to NIR) can be logically prepared. Did the authors try this?

Reviewer #3:

Remarks to the Author:

The authors claim to report first triangular carbon qdots with both narrow and size dependent emission lines. The size dependence of emission is correlated with structural characterisation, revealing flat, sp² coordinated graphene qdots with size variation from 1.9 to 3.9 nm and resulting red shift of emission. The nature of the edges and functionalization however is not so clear. With about 4nm size the dots contain about few hundred atoms. The authors support their analysis with DFT calculations of a number of very small carbon molecules. I agree with the authors that a convincing demonstration of narrow emission with systematic size dependence of these graphene quantum dots would be really exciting. However, contrary to authors claim, triangular graphene quantum dots have been already demonstrated, with structural properties almost identical to a number of structures in Fig.4. These structural properties were related to characteristic excitonic states in a number of papers, e.g., Ozfidan et al. Phys.Rev.B89,085310 (2014). Absorption, emission and time resolved measurements very similar to Fig.2 have been reported in .e.g. Sun et al. NanoLetters 15,5742(2015). The size dependence of the electronic and optical properties of graphene qdots were described by Guclu et al in a monograph by Springer Verlag "Graphene quantum dots" and by others. There is also nice work by Stampfer and Morgenster using STM. We also point out that optical properties of forms of carbon, from intercalated graphite to carbon nanotubes have been studied and demonstrated, with Mildred Dresselhaus as one of the names which comes to mind. It is therefore up to the authors to properly review the state of the knowledge and convince the reader, and the referee, about the novelty of their work. If this is done convincingly I will be pleased to consider this paper for publication.

Responselies to Reviewers' Comments and Revisions Made

~~(NCOMMS-18-00184)~~

Reviewer 1

In this manuscript (NCOMMS-18-00184), the authors suggested a very interesting synthetic idea and route to form stable crystalline structure and high quality triangular shaped CQDs which show both high color-purity with smaller FWHM in PL and higher thermostability than ever reported. It was also revealed that T-CQDs with electron-donor function group –OH are most efficient compared to those with electron accepting groups like COOH, C=O etc by reducing electron-phonon scattering through high delocalization. T-CQDs-based LED show highly bright luminance and current efficiency when PVK polymer was incorporated as host material.

This paper will be appropriate for publication in NCOMMs, but the following questions should be further clarified before acceptance.

Reply: We are very appreciative of the Reviewer's high evaluations in different aspects as well as the constructive and valuable comments, which have been addressed carefully point by point as below.

1. What's the exact name of fluorescein used as a comparative reference in S26?

Reply: We thank the Reviewer for the suggestion. The exact name of fluorescein is 2-(6-hydroxy-3-oxo-3h-xanthen-9-yl)-benzoic acid.

Accordingly, we have added the exact name of fluorescein when discussing about the stability of NBE-T-CQDs in the optical properties section in our manuscript as follows:

"Moreover, besides the high thermostability concluded from the temperature-dependent PL spectra, the NBE-T-CQDs also showed more robust photostability than the best protected core-shell inorganic QDs such as CdZnS@ZnS and organic dyes such as **fluorescein (2-(6-hydroxy-3-oxo-3h-xanthen-9-yl)-benzoicaci)** under continuous radiation of a UV lamp for 10 h (Supplementary Figure S27), giving them more competitive edges for LED applications."

2. Exciton binding energy of ca. 139 meV for B-NBE-T-CQD decreased to that of ca.100 meV of R-NBE-T-CQDs. Could explain why the exciton binding energies of T-CQDs were quite larger than any other inorganic (e.g. 60 meV for ZnO) and were reduced as the emission center is changed from blue to red?

Reply: We appreciate the Reviewer's acute question. Exciton binding energy is one of the most crucial physical parameters for optoelectronic materials. The values of exciton binding energies are usually closely related with the dimensionality, size, crystallinity, and surface defects of materials. As for bulk semiconductors, due to the effective Coulomb screening effect with very weak bound excitons, the values of the exciton binding energies are only a few tens of meV (such as 60 meV for ZnO and 25 meV for GaN). With the reduction of the dimension, the less Coulomb screening effect will give rise to more strongly bound excitons and thus result in larger exciton binding energies. The linear relationship of the exciton binding energy versus the bandgap of low dimensional materials such as graphene fluoride has also been demonstrated, and the exciton binding energy increased to be as high as 2030 meV when the bandgap was about 7.7 eV.

To the best of our knowledge, this is the first time that we obtained the important physical parameter of exciton binding energy for the carbon quantum dots. We deduce that such a large exciton binding energy of ca. 139 meV for B-NBE-T-CQD should be due to confinement effect associated with its unique quantum-sized high crystalline triangular graphene structure, and it decreases from 139.2 (B-NBE-T-CQDs) to 100.6 meV (R-NBE-T-CQDs) in a way conforming to the

decreased bandgap from blue to red NBE-T-CQDs, again due to the confinement effect involving the size-dependent coulomb interaction. Of course, much more research about the exciton binding energy of CQDs will be further deeply investigated in the near future.

Accordingly, we have added the sentences in our manuscript as following:

"The relatively large exciton binding energy of NBE-T-CQDs decreased from 139.2 (B-NBE-T-CQDs) to 100.6 meV (R-NBE-T-CQDs) in a way conforming to the corresponding bandgap decrease, which is probably due to confinement effect associated with the unique quantum-sized high crystalline triangular graphene structure involving the size-dependent coulomb interaction⁴⁶ and bandgap energy⁴⁷."

3. In Fig. 4, how can the degrees of delocalization be defined in HOMO and LUMO levels for T-CQDs-OH, T-CQDs, T-CQDs-COOH, and S-CQDs by DFT calculation?

Reply: Thanks for bringing up the question. The degree of delocalization of the HOMO and LUMO can be qualitatively judged, for instance, by their corresponding electron cloud density distributions around the whole molecular structure. In this sense, a more uniformly distributed electron cloud density would indicate a higher degree of delocalization. For example, the calculated electron cloud densities of T-CQDs-3 appear to be rather uniformly distributed across the whole molecular structure as shown in **Figure R1a, b**, whereas that of T-CQDs-COOH-3 are only accumulated in some specific areas of the molecular structure as shown in **Figure R1c, d**. Therefore, it can be concluded that the HOMO and LUMO of T-CQDs-3 show higher degrees of delocalization than those of T-CQDs-COOH-3.

Figure R1. The calculated HOMO and LUMO molecular orbitals of T-CQDs-3 (a, b) and T-CQDs-COOH-3 (c, d).

Accordingly, we have added more detailed discussion about the degrees of delocalization defined from calculated HOMO and LUMO molecular orbitals in the theoretical investigation section in our manuscript as following:

"The degree of delocalization of the HOMO and LUMO can be qualitatively judged, for instance, by their corresponding electron cloud density distributions around the whole molecular structure. In this sense, a more uniformly distributed electron cloud density would indicate a higher degree of delocalization. Clearly, the calculated electron cloud densities of HOMO and LUMO for both T-CQDs-OH and T-CQDs are more uniformly distributed across the whole molecular structure than those for T-CQDs-COOH and S-CQDs (Figure 4m-x, Supplementary Figure S46-52). Therefore, it can be reasonably concluded that the HOMOs and LUMOs of T-CQDs-OH and T-CQDs show higher degrees of delocalization than those of T-CQDs-COOH and S-CQDs."

And therefore how can they connect with the FWHM of PL spectra?

Reply: The degrees of delocalization of CQDs are closely connected with the FWHM of PL spectra. Higher degrees of delocalization of CQDs could lead to higher structural stability, which has also been demonstrated by theoretical calculations as shown in Table S15 in Supplementary information. Increased structural stability can result in dramatically reduced electron-phonon coupling, and then lead to the high color-purity excitonic emission and narrow FWHM of PL spectra of NBE-T-CQDs as demonstrated by the temperature-dependent PL spectra.

Accordingly, in the theoretical investigation section in our manuscript, we have added more detailed discussion about the connection between the degrees of delocalization and FWHM of PL spectra as following:

"Interestingly, the higher degree of delocalization leads to higher structural stability of the unique triangular structure of the NBE-T-CQDs, which in turn results in dramatically reduced electron-phonon coupling. This contributes to the high color-purity excitonic emission and narrow FWHM of PL spectra of NBE-T-CQDs as demonstrated by the temperature-dependent PL spectra."

What does the pink and violet colors of each (a), (b), (d) and (e) in S51-55 mean? Please add the explanation in figure caption or in text.

Reply: We thank the Reviewer's question and suggestion. The pink and violet colors in the HOMO and LUMO molecular orbitals represent the positive and negative phase of the molecular orbital wavefunctions.

Accordingly, we have added relative description about the pink and violet colors in the HOMO and LUMO molecular orbitals in the theoretical calculations method section in our manuscript as following:

"The pink and violet colors in the HOMO and LUMO molecular orbitals represent the positive and negative phases of the molecular orbital wavefunctions."

4. As the ref. [18], it will be also important to estimate the luminance, current efficiency of the pure (B, G, Y, G)-T-CQDs-OH based LED without being mixed with PVK polymer host. Then it should be compared how much it was improved when PVK host was used.

Reply: According to the Reviewer's suggestion, we have measured the luminance and current efficiency of B-, G-, Y-, and R-LEDs based on pure NBE-T-CQDs without being mixed with PVK polymer host, and added the results as Table S17 in supplementary information.

Accordingly, we have also added more detailed discussion about the performance comparison of the NBE-T-CQDs-based LEDs with and without using PVK polymer host in LED performance section in our manuscript as following:

"Compared with the NBE-T-CQDs-based LEDs fabricated without using the PVK polymer host, the one with PVK exhibited greatly improved L_{max} and η_c by 12-25 and 17-28 times, respectively, as shown in Table S17. Besides the bright fluorescence inherent to the NBE-T-CQDs, the hole-transport PVK polymer as a host material also contributed to the remarkable performance of our LEDs due to the resulting optimized charge balance in the emission layer."

LEDs	L_{max} (cd m ⁻²)	η_c (cd A ⁻¹)	L_{max} (cd m ⁻²)	η_c (cd A ⁻¹)
B-LEDs	1882	1.22	162	0.07
G-LEDs	4762	5.11	260	0.35
Y-LEDs	2784	2.31	126	0.09
R-LEDs	2344	1.73	92	0.06

Table S17. Performance comparison of the NBE-T-CQDs-based LEDs fabricated with and without using PVK polymer host.

Reviewer 2

The exploitation of new method for engineering high quality carbon quantum dots (CQDs) with multicolor and narrow bandwidth emission is very challenging and desired. In this manuscript, the authors reported preparation methods to achieve unprecedented triangular CQDs by a facile solvent thermal process. Meanwhile, the as-prepared CQDs exhibit tunable multicolor fluorescence emission with very narrow full-width-at-half-maximum (FWHM) (~30 nm) and high absolute quantum yield (up to 72%). The unique triangular structures of CQDs were fully characterized by TEM, Raman, XRD, NMR and XPS, especially high-angle annular dark-field scanning transmission electron microscopy (HAADF-STEM). In addition, the formation mechanism for the achievement of the triangular CQDs was also discussed. Based on the unprecedented structure and unique optical properties of the as-prepared CQDs, multicolored LEDs with high color-purity and stability renders this manuscript to be acceptable by Nature Communications, if the following questions can be addressed.

Reply: We thank the Reviewer for the very high evaluation on our work. We have addressed the questions carefully point by point as below.

1. The reproducibility of preparation has to be confirmed and provided! We had followed the procedures that the authors described; unfortunately, we cannot obtain the same results as that of the authors reported in this manuscript. I suggest the authors to provide third-party results to prove the reproducibility of their method.

Reply: We are very appreciative of the question about the reproducibility and indeed very glad to know the interest of the Reviewer's lab in following the experiments. We are also very sorry to hear that He/She did not obtain the same results. We suspect that some experimental details such as the detailed purification processes might have been ignored in the preparation procedures. To be more helpful in this regard, we have added more experimental details about the purification process in the preparation procedures in method section in our manuscript as following:

"The NBE-T-CQDs are purified via silica column chromatography using a mixture of dichloromethane and methanol as the eluent. During the silica column chromatography purification process, the polarity of eluent should be changed dynamically by changing the volume ratio of dichloromethane to methanol in order to ensure the effectiveness of the separation and purification process. Specifically, the volume ratio was dynamically changed during the silica column chromatography purification process from 6:1 to 2:1 (blue NBE-T-CQDs), 10:1 to 4:1 (green NBE-T-CQDs), 16:1 to 8:1 (green NBE-T-CQDs), and 25:1 to 10:1 (red NBE-T-CQDs). Typically, the silica column chromatography purification process should be repeated several times in order to obtain pure NBE-T-CQDs."

Anyway, our results are beyond any doubt. In fact, several students in our lab are working in parallel on the preparation of high color-purity narrow bandwidth emission CQDs as well as their applications for LEDs. For instance, our recent success on such synthesis is the first example of efficient and stable white LEDs (WLEDs) based on narrow bandwidth emission CQDs for wide-color-gamut backlight displays as shown in Figure R2. The WLEDs shows very stable and efficient white light with a power efficiency of 86.5 lm W^{-1} with the color coordinate of (0.35,0.33) and a wide color gamut of 110% NTSC (National Television System Committee), even superior to that of previously reported semiconductor QDs-based WLEDs. This work will be reported in the near future.

Figure R2. The EL spectra (a) and corresponding color gamut (white line) in CIE 1931 diagram (b) of WLEDs fabricated based on narrow bandwidth emission CQDs.

Nonetheless, according to the Reviewer's suggestion, we have provided the third-party results to prove the reproducibility of our method for the preparation of high color-purity narrow bandwidth emission CQDs as attached file below. In addition, a short video has also been provided to describe the preparation, purification and the PL spectra measurement process.

Monday, March 12, 2018

To whom it may concern,

It is my great pleasure to write a supporting letter for the manuscript of NCOMMS-18-00184. On March 2, 2018, I was invited to check the reproducibility of synthetic method reported in the manuscript for the preparation of high color-purity narrow bandwidth emission carbon quantum dots (CQDs). I witnessed the fabrication process including synthesis, purification and photoluminescence (PL) measurements by Mr. Fanglong Yuan and Ms Zifan Xi. The PL spectra of these samples are consistent with that reported in the manuscript (see figure 1). Based the above results, I would like to confirm that the method reported in the manuscript can be well reproduced.

Figure 1. The PL spectra obtained on the spot

Yours sincerely

北京理工大学
BEIJING INSTITUTE OF TECHNOLOGY

Haizheng Zhong
Professor of Photonic Materials
School of Materials Science & Engineering
Beijing Institute of Technology
5 Zhongguancun South Street, Beijing, China 100081
Phone: +86-10-68918188, Fax: +86-10-68918188
E-mail: hzzhong@bit.edu.cn
<http://teacher.bit.edu.cn/hzzhong/yjcg/index.htm>

中国·北京市·海淀区中关村南大街五号 邮政编码: 100081 <http://www.bit.edu.cn>
5 South Zhongguancun Street, Haidian, Beijing 100081, P. R. China Tel: (86-10) 68918188

2. The preparation yield for the materials should be provided.

Reply: Thanks for the Reviewer's suggestion. The preparation yield for NBE-T-CQDs is estimated to be about 8-13%.

Accordingly, we have added the relative description about the preparation yield in the synthesis and optical properties section in our manuscript as following:

"The preparation yield for NBE-T-CQDs is estimated to be about 8-13%."

3. The relatively low resolution TEM image should also be provided, which can be used to illustrate the general morphologies of the as-prepared materials. In addition, AFM is also better to measure.

Reply: In fact, the relatively wide-area TEM images have been obtained as shown in Figure S28, which clearly demonstrated a narrow size distribution of the nanoparticles of CQDs with the obvious high crystalline triangular structure. The relative description about the relatively wide-area TEM images in structural characterizations section in our manuscript are as follows:

"The wide-area TEM images of NBE-T-CQDs all show a narrow size distribution of the nanoparticles with the distinctive high crystalline triangular structure as highlighted by the white contour lines (Figure 3b, Supplementary Figure S28)."

Figure S28. TEM images and the corresponding size distribution of B- (a,b), G- (c,d), Y- (e,f), and R-NBE-T-CQDs (g,h).

As for the TEM image obtained at a much lower resolution, because of the relatively thin thickness and low contrast, when the samples were separated and dispersed well, the TEM images like the below Figure R3 could be obtained. The resolution of AFM

image is also relatively too lower to clearly demonstrate the morphologies of CQDs when the samples were separated and dispersed well.

Figure R3. The TEM image of G-NBE-T-CQDs obtained at lower resolution.

4. The wavelength value should be added in the x-axis in Figure 1h and 1i;

Reply: We thank the Reviewer for the suggestion. The UV-vis absorption and PL full spectra of NBE-T-CQDs with wavelength value in the x-axis has already been provided as shown in Figure S1 and S2.

Figure S1. Normalized UV-vis absorption (a) and PL (b) spectra of B-, G-, Y-, and R-NBE-T-CQDs.

Figure S2. Normalized UV-vis absorption and PL spectra of B- (a), G- (b), Y- (c), and R-NBE-T-CQDs (d).

To avoid the overlap of the UV-vis absorption and PL full spectra of NBE-T-CQDs in a single figure, we shifted the absorption and PL spectra in Figure 1h and 1i along the x-axis to clearly display the spectral characteristics, and the absorption and PL peaks were also clearly indicated.

the FL excitation spectra of the prepared CQDs should be measured and provided.

Reply: According to the Reviewer's suggestion, we have added the results of the FL excitation spectra of B-, G-, Y-, and R-NBE-T-CQDs as Figure S3 in supplementary information.

Accordingly, we have added the relative description about the FL excitation spectra of B-, G-, Y-, and R-NBE-T-CQDs in the synthesis and optical properties section in our manuscript as following:

"The maximum peak wavelength of the FL excitation spectra are centered at about 460 (B-), 498 (G-), 521 (Y-) and 582 nm (R-NBE-T-CQDs), and agree well with the corresponding excitonic absorption peak wavelengths (Supplementary Figure S3-4), clearly suggesting that the emission of NBE-T-CQDs originates from band-edge exciton-state decay rather than from defect-state decay."

Figure S3. The FL excitation spectra of B- (a), G- (b), Y- (c), and R-NBE-T-CQDs (d).

5. In Figure 1f, is the fluorescence image really obtained under the same single UV excitation wavelength (365 nm)? This seems does not reasonable, because no absorption of the G-, Y-, and R-CQDs at 365 nm!

Reply: We are very sorry for causing the misunderstanding of "no absorption of the G-, Y-, and R-CQDs at 365 nm". As we have discussed above, the absorption and PL spectra in Figure 1h and 1i have been shifted along the x-axis in order to clearly display the spectra characteristics. In fact, the fluorescence image is really obtained with the same single UV excitation wavelength (365 nm), and as shown in Figure S2, the G-, Y-, and R-CQDs all show obvious absorption at 365 nm in the absorption spectra.

Figure S2. Normalized UV-vis absorption and PL spectra of B- (a), G- (b), Y- (c), and R-NBE-T-CQDs (d).

Accordingly, we have added relative description about the absorption of NBE-T-CQDs at 365 nm to avoid the misunderstanding in the synthesis and optical properties section in our manuscript as following:

"And the emission colors are brighter under UV light irradiation (excited at 365 nm, Figure 1g) due to the obvious absorption of NBE-T-CQDs at 365 nm (Supplementary Figure S1-2)."

6. Please indicate the emission and excitation wavelengths of the decay spectra in the time-resolved decay curve measurement in Figure 2a.

Reply: Thanks for the suggestion. The emission and excitation wavelengths of the decay spectra in the time-resolved decay curve measurement in Figure 2a are about 472/460, 507/500, 538/520, and 598/580 nm for B-, G-, Y-, and R-NBE-T-CQDs, respectively.

Accordingly, we have added relative description about the emission and excitation wavelengths of the decay spectra in the time-resolved decay curve measurement in Figure 2a in the synthesis and optical properties section in our manuscript as following:

"To gain more insight into the exciton recombination dynamics, we measured time-resolved PL spectra with emission and excitation wavelengths of about 472/460, 507/500, 538/520, and 598/580 nm for B-, G-, Y-, and R-NBE-T-CQDs, respectively, and the results are shown in Figure 2a."

7. Based on the reaction conditions and supposed reaction mechanism, CQDs with longer emission wavelengths (e.g., pure red to NIR) can be logically prepared. Did the authors try this?

Reply: This is a good question! In fact, up to now, we have tried hard but failed to prepare high color-purity narrow bandwidth emission CQDs with longer emission wavelengths (> 610 nm) by changing the reaction conditions with phloroglucinol as precursor. It is likely that as the reaction goes on, the reactivity of the system may become lower and further reaction to produce larger size of CQDs would become harder. However, the preparation of CQDs with longer emission wavelengths (e.g., pure red to NIR) with phloroglucinol as precursor is still under active exploration in our lab.

In addition, we have recently prepared high color-purity narrow bandwidth emission CQDs with emission peak at 665 nm and FWHM of 48 nm (Figure R4) by changing the precursor of phloroglucinol to other precursors, which will be reported in due course.

Figure R4. The high color-purity narrow bandwidth emission CQDs with emission peak at 665 nm and FWHM of 48 nm.

Reviewer 3

The authors claim to report first triangular carbon qdots with both narrow and size dependent emission lines. The size dependence of emission is correlated with structural characterisation, revealing flat, sp^2 coordinated graphene qdots with size variation from 1.9 to 3.9 nm and resulting red shift of emission. The nature of the edges and functionalization however is not so clear. With about 4nm size the dots contain about few hundred atoms. The authors support their analysis with DFT calculations of a number of very small carbon molecules. I agree with the authors that a convincing demonstration of narrow emission with systematic size dependence of these graphene quantum dots would be really exciting.

Reply: We are very appreciative of the Reviewer's high evaluation. We are also very glad to learn of the Reviewer's recognition "demonstration of narrow emission with systematic size dependence of these graphene quantum dots would be really exciting."

To be sure, our work reported in the paper has clearly presented a convincing demonstration of narrow emission with systematic size dependence of these graphene quantum dots. We will emphasize the following points to address this issue:

1) As the size of NBE-T-CQDs increased from 1.9 to 3.9 nm, the corresponding narrow bandwidth emission peak also red-shifted from 472 nm (blue) to 598 nm (red), which clearly and directly demonstrated an almost linear size dependent narrow bandwidth emission property as shown in Figure S6.

Figure S6. Dependence of the PL and the first excitonic absorption peak wavelength on the size of NBE-T-CQDs.

In fact, the systematic size dependent narrow bandwidth emission of NBE-T-CQDs has already been discussed in the synthesis and optical properties section in our manuscript as following:

"The gradually red-shifted narrow bandwidth excitonic emission peak of the NBE-T-CQDs from 472 nm (blue) to 598 nm (red) are well consistent with the corresponding increased size from 1.9 to 3.9 nm. Significantly, such a correlation is almost linear as shown in Supplementary Figure S6, a very clear characteristic of the bandgap transitions of the NBE-T-CQDs."

2) The theoretical investigation in our work has also demonstrated that the unique highly crystalline triangular CQDs structure show a size dependent narrow bandwidth emission property. As the number of fused benzene rings of triangular CQDs increased from 4 to 19, the calculated narrow bandwidth emission also red-shifted from 323 to 491 nm (Table S14, more detailed discussions can be found in Page 5-6 in the manuscript).

Table S14. The calculated emission wavelength (nm), FWHM (nm), HOMO (eV), LUMO (eV) and band gap energies (eV) of the different kinds of model CQDs.

	S-CQDs			T-CQDs			T-CQDs-OH			T-CQDs-COOH		
	1	2	3	1	2	3	1	2	3	1	2	3
Em	398	496	524	312	411	481	323	430	491	354	434	500
FWHM	85	132	149	52	74	77	48	62	64	68	103	135
HOMO	5.39	5.16	5.06	5.96	5.49	5.27	5.34	5.25	5.07	6.85	6.30	5.65
LUMO	1.95	2.25	2.40	1.38	1.75	2.19	1.26	1.72	2.11	2.98	2.88	2.78
GAP	3.44	2.91	2.66	4.58	3.74	3.08	4.08	3.53	2.96	3.87	3.42	2.87

The nature of the edges and functionalization of CQDs in our work has been clearly demonstrated by numerous characterization techniques. The DFT calculated optical properties of triangular model CQDs are also well consistent with the experimental results in our work, which has been discussed in detail in the theoretical investigation section in our manuscript.

However, contrary to authors claim, triangular graphene quantum dots have been already demonstrated, with structural properties almost identical to a number of structures in Fig.4. These structural properties were related to characteristic excitonic states in a number of papers ,e.g.. Ozfidan et al. Phys.Rev.B89,085310 (2014).

Reply: We thank the Reviewer for bringing to our attention the paper (Ozfidan et al. Phys. Rev. B), and we also agree that the paper has **theoretically revealed** that the structural properties of graphene quantum dot are related to the characteristic excitonic states. However, we want to emphasize the following points to demonstrate that the main point of this paper (Ozfidan et al. Phys. Rev. B) is totally different from ours in this work:

1) This paper (Ozfidan et al. Phys. Rev. B) revealed that the structural properties of graphene quantum dot were related to characteristic excitonic states **from a theoretical point of view**. However, in our work, we **experimentally** demonstrate the first synthesis and optical properties of novel different sized triangular CQDs with

defect-free graphene crystalline structure, the structure of which has been clearly revealed by HAADF-STEM images for the first time (Figure R5).

Figure R5. The typical aberration-corrected HAADF-STEM images of B- (a), G- (b), Y- (c), and R-NBE-T-CQDs (d), respectively. Scale bar, 2 nm.

2) The detailed content of theoretical research of this paper (Ozfidan et al. Phys. Rev. B) is also quite different from our current work. This paper (Ozfidan et al. Phys. Rev. B) mainly presented a microscopic theory of biexcitons in graphene quantum dots by using the tight-binding (TB) method and self-consistent-Hartree-Fock (HF) method. But in our work, we mainly investigated the optical properties (including the PL spectra, FWHM, HOMO and LUMO energy levels, and bandgap energies) of different kinds of model CQDs by using the time-dependent density functional theory method as implemented in the Gaussian09 software package.

In addition, we have cited this paper (Ozfidan et al. Phys. Rev. B) as reference 49 in the theoretical investigation section in our manuscript.

Absorption, emission and time resolved measurements very similar to Fig.2 have been reported in .e.g. Sun et al. NanoLetters 15,5742(2015).

Reply: Again thanks for bringing the paper to our attention. However, the TA spectra in the work of (Sun, Figge et al. Nano Letters 15,5742, 2015) (Figure R6a) are quite different from the fs-TA spectra of NBE-T-CQDs in our work (Figure R6b) in following ways:

The TA spectra in the work of (Sun, Figge et al. Nano Letters 15,5742, 2015) only show ground state bleaching and excited state absorption signals with scan delay time limited to be 100 ps (Figure R6a), but without observing the signal of

stimulated emission. On the contrary, apart from the observed signals of ground state bleaching and excited state absorption in the fs-TA spectra with long scan delay time to be 2000 ps, the high color-purity narrow bandwidth stimulated emission of NBE-T-CQDs is also observed for the first time experimentally in our work (Figure R6b). This is a fundamental difference, which has been both experimentally and theoretically demonstrated to be originated from the unique quantum-sized high crystalline triangular graphene structure with almost no defects synthesized in our work.

We agree that that intensive theoretical work has been conducted on the electronic and optical properties of colloidal graphene quantum dots, and great efforts have also been made to improve the optical properties. However, it still remains a widely accepted belief that CQDs can give only BROAD emission and inferior color-purity with full width at half maximum (FWHM) commonly exceeding 80 nm. In our work, we have designed novel triangular CQDs that can deliver light emission (from blue to red) with an unprecedented narrow bandwidth of 29 nm for the first time. This has changed the way we think about CQDs LEDs.

Figure R6. The TA spectra in the literature (Sun, Figge et al. Nano Letters 15,5742, 2015) (a) and the fs-TA spectra of NBE-T-CQDs in our work (b).

The size dependence of the electronic and optical properties of graphene qdots were described by Guclu et al in a monograph by Springer Verlag “Graphene quantum dots” and by others.

Reply: We agree with the Reviewer that the size dependence of the electronic and optical properties of graphene qdots were already described, which indeed has been

widely known for many years. However, the reported size dependent PL spectra from graphene qdots all shows BROAD bandwidth with FWHM larger than 80 nm. On the contrary, in our work, it is the first time that we demonstrate the novel quantum-sized high crystalline triangular CQDs with an almost linear size dependent narrow bandwidth (FWHM of 29 nm) emission (from blue to red) property as shown in Figure S6, which is fundamentally different from previously reported literatures.

Figure S6. Dependence of the PL and the first excitonic absorption peak wavelength on the size of NBE-T-CQDs.

In addition, we have cited this monograph (Guclu et al. Springer Verlag “Graphene quantum dots”) as reference 50 in the theoretical investigation section in our manuscript.

There is also nice work by Stampfer and Morgenster using STM.

Reply: We also agree with the Reviewer that this paper (Stampfer, Appl. Phys. Lett. 103, 111604 (2013)) has performed STM on micron-sized graphene grown on SiC, but it is not directly related to our work. In our work, we emphasize the first demonstration of the synthesis of novel triangular CQDs with defect-free graphene crystalline structure to deliver light emission (from blue to red) with ultra-narrow bandwidth as well as the demonstration of the triangular graphene crystalline structure by HAADF-STEM images for the first time (Figure R5).

We also point out that optical properties of forms of carbon, from intercalated graphite to carbon nanotubes have been studied and demonstrated, with Mildred Dresselhaus as one of the names which comes to mind.

Reply: We agree with the Reviewer that the optical properties of forms of carbon, from intercalated graphite to carbon nanotubes have been studied and demonstrated extensively. However, up to now, the reported PL spectra from different kinds of carbon materials all shows BROAD bandwidth with FWHM larger than 80 nm, which are totally different from ours in this work. We want to emphasize again that one of main novelties in our work is that we demonstrate the first synthesis of novel triangular CQDs with defect-free graphene crystalline structure to deliver light emission (from blue to red) with narrow bandwidth of 29 nm.

It is therefore up to the authors to properly review the state of the knowledge and convince the reader, and the referee, about the novelty of their work. If this is done convincingly I will be pleased to consider this paper for publication.

Reply: We thank the Reviewer for the suggestion. Actually, we have also carefully read and studied previous literatures on carbon quantum dots extensively before. According to the Reviewer's suggestion, we have added more detailed discussion to review the state of the knowledge of CQDs in introduction section in our manuscript as following:

"However, despite the intensive work on the electronic and optical properties of CQDs, it has until now remained a widely accepted belief that CQDs can only give broad emission and inferior color-purity with full width at half maximum (FWHM) commonly exceeding 80 nm^{15-19,23}."

After careful comparison with existing experimental and theoretical results on previously reported graphene quantum dots, we will emphasize the following points to highlight the true originality and novelty of our work:

1) For the first time, we report the synthesis of novel triangular CQDs with defect-free graphene crystalline structure, a brand new carbon nanostructure, which has been clearly revealed by HAADF-STEM images (Figure R5). To the best of our knowledge, this is the first time that the exquisite aberration-corrected HAADF-STEM images of carbon materials are obtained.

Figure R5. The typical aberration-corrected HAADF-STEM images of B- (a), G- (b), Y- (c), and R-NBE-T-CQDs (d), respectively. Scale bar, 2 nm.

2) With the novel high crystalline triangular CQDs, we are able to realize high color-purity, narrow bandwidth (FWHM of 29 nm) and multicolored (from blue to red) emission with a quantum yield up to 72% (Figure 1f-i).

Figure 1f-i. Photographs of the NBE-T-CQDs ethanol solution under daylight (f) and fluorescence images under UV light (excited at 365 nm) (g). The normalized UV-vis absorption (h) and PL (i) spectra of B-, G-, Y-, and R-NBE-T-CQDs, respectively.

3) The in-depth studies of fs-TA spectroscopy and temperature-dependent emission narrowing were first conducted to reveal the emission mechanism of NBE-T-CQDs, which demonstrated the dramatically reduced electron-phonon coupling due to the

triangular structural rigidity as the origin of the high color-purity excitonic emission (Figure 2).

Figure 2. Ultrafast dynamics of the photoexcited states and temperature-dependent PL spectra of the NBE-T-CQDs.

4) Significantly, the very important physical parameters of exciton binding energy of carbon materials were obtained for the first time.

5) The emission mechanism of NBE-T-CQDs was also systematically studied by elaborate theoretical calculations. It was revealed that the unique triangular CQDs showed highly delocalized charges and high structural stability, which subsequent resulted in dramatically reduced electron-phonon coupling and further led to the high color-purity excitonic emission.

6) For the first time, based on the NBE-T-CQDs, high-performance and high color-purity multicolored electroluminescent LEDs (FWHM of 30 nm) with a maximum luminance of 4762 cd/m^2 and current efficiency of 5.11 cd/A have been realized (Figure 5), which was even superior to the well-developed semiconductor QDs-based LEDs. Also demonstrated is the outstanding stability of such LEDs both on shelf and in operation. This work will set the stage for developing high color-purity and high-performance NBE-T-CQDs-based LEDs ideal for the next-generation display technology.

Figure 5. LED structure, energy diagram and performance characterization.

To summarize, our work has allowed the CQDs to climb up to the rank of inorganic QDs such as CdSe QDs in terms of light-emitting performance. Underlying the breakthrough lies the triangular CQDs, which are fundamentally transformative and turn the traditional conception on its head that carbon QDs can give only broad emission and inferior color-purity.

Reviewers' Comments:

Reviewer #1:

Remarks to the Author:

As a reviewer 1, I think that all the claimed questionnaires about the exact name of fluorescein, degree of delocalization and related FWHM of PL, the origin of high exciton binding energy, and the measurement of EL for pure (B, G, Y, G)-T-CQDs-OH based LED are reasonably well responded and revised point-by-point The authors' explanation, I believe, are quite acceptable.

I recommend the revised manuscript is quite acceptable in this form to Nature Comm.

Reviewer #2:

Remarks to the Author:

The authors had properly responded my concerns, some other minor suggestions:

1) The excitation wavelengths and solvents should be given for the PL emission quantum yields (page 3, top of the right column);

2) How about the solid (powder) PL emission properties of the synthesized materials?

3) In the abstract and conclusion sections, the authors only offered the "best data" (e.g., narrow FWHM 29 nm, PL quantum yield 72%, LED performance L_{max} 4762 cd/m² and η_c 5.11 cd/A, and etc.), and this seems not good. I suggest ranges (rather than only the best data) are better to offer.

4) In the theoretical investigation section, the models were set containing only several benzene ring units, does this appropriate? Since the synthesized materials are several nanometers, which contain much more benzene units.

Reviewer #3:

Remarks to the Author:

The Authors unfortunately did not take my comments seriously. There are several papers, experimental and theoretical, which describe optical properties of triangular graphene quantum dots (TGQDs). These dots have exactly either 168 or 132 carbon atoms hence the size distribution is ZERO. The absorption and emission spectra were measured, including time resolved absorption, with one of most recent measurements reported in Sun, Figge et al. Nano Letters 15,5742, 2015). One can easily trace all experiments on these TGQDs from this paper in NanoLetters. The theory of optical properties of TGQDs was given in a paper by Ozfidan et al. Phys.Rev.B89,085310 (2014). The Authors decided to ignore these references. They continue to introduce QDs using terms like "it is widely believed" and portraying the triangular shape of graphene dots as their major innovation. This is not correct. The Authors must start their paper by analysing state of knowledge, i.e., all existing experiments and theory of TGQDS with exactly known number of carbon atoms and their measured optical properties including excitonic and bi-excitonic states, and focus on convincing the reader what is the novelty of their work. Otherwise, one must conclude that since the triangular graphene quantum dots with exactly known structure were already made, measured and theoretically analysed, there is not enough novelty to warrant publication in NatComm.

Response to Reviewers' Comments and Revisions Made

Reviewer 1

As a reviewer 1, I think that all the claimed questionnaires about the exact name of fluorescein, degree of delocalization and related FWHM of PL, the origin of high exciton binding energy, and the measurement of EL for pure (B, G, Y, G)-T-CQDs-OH based LED are reasonably well responded and revised point-by-point. The authors' explanation, I believe, are quite acceptable. I recommend the revised manuscript is quite acceptable in this form to Nature Comm.

Reply: We are very appreciative of the Reviewer's high evaluations of our revisions, and we also thank the Reviewer for recommending the acceptance of our manuscript.

Reviewer 2

The authors had properly responded my concerns, some other minor suggestions:

Reply: We thank the Reviewer for the high evaluation of our revisions. We have addressed the minor suggestions carefully point by point as below.

1. The excitation wavelengths and solvents should be given for the PL emission quantum yields (page 3, top of the right column).

Reply: We thank the Reviewer for the suggestion. The PL emission quantum yields were determined in ethanol with corresponding excitation wavelengths of 460, 500, 520 and 580 nm for B-, G-, Y-, and R-NBE-T-CQDs, respectively.

Accordingly, we have added the excitation wavelengths and solvent when discussing about the quantum yield of NBE-T-CQDs in our manuscript as following:

"The absolute QY was determined to be 66%, 72%, 62%, 54% in ethanol for the high color-purity B-, G-, Y-, and R-NBE-T-CQDs with the corresponding excitation wavelength being 460, 500, 520 and 580 nm, respectively."

2. How about the solid (powder) PL emission properties of the synthesized materials.

Reply: We thank the Reviewer for the question. The NBE-T-CQDs shows almost no obvious fluorescence in the macroscopic solid powder state due to the strong π - π interactions between the high crystalline NBE-T-CQDs with a large π conjugated structure. In fact, this is a quite common characteristic and phenomenon that almost no fluorescence can be observed in macroscopic solid powder state CQDs.

Accordingly, we have added relative description about the solid (powder) PL emission properties of NBE-T-CQDs in our manuscript as following:

"Almost no fluorescence was detected in the macroscopic solid powder state due to the strong π - π interactions between the highly crystalline NBE-T-CQDs with a large π conjugated structure."

3. In the abstract and conclusion sections, the authors only offered the "best data" (e.g., narrow FWHM 29 nm, PL quantum yield 72%, LED performance L_{\max} 4762 cd/m^2 and η_c 5.11 cd/A , and etc.), and this seems not good. I suggest ranges (rather than only the best data) are better to offer.

Reply: We thank the Reviewer for the suggestion. Accordingly, we have changed the best data to the ranges in the abstract and conclusion sections as following:

In the abstract section:

"Here we demonstrate high color-purity, narrow bandwidth (FWHM of **29-30** nm) and multicolored (from blue to red) emission from specially designed triangular CQDs (T-CQDs), with a quantum yield up to **54-72%**."

"Moreover, multicolored electroluminescent light-emitting diodes (LEDs) based on the narrow bandwidth emission T-CQDs (NBE-T-CQDs) displayed high color-purity (FWHM of **30-39** nm) as well as high-performance with a maximum luminance of **1882-4762** cd/m² and current efficiency of **1.22-5.11** cd/A."

In the conclusion section:

"We report the subversive demonstration of high color-purity NBE-T-CQDs (FWHM of **29-30** nm) from blue to red with a QY up to **54-72%**."

"The multicolored LEDs based on the NBE-T-CQDs demonstrated high color-purity (FWHM of **30-39** nm), a L_{\max} of **1882-4762** cd/m² and η_c of **1.22-5.11** cd/A."

4. In the theoretical investigation section, the models were set containing only several benzene ring units, does this appropriate? Since the synthesized materials are several nanometers, which contain much more benzene units.

Reply: We thank the Reviewer for the question. In the theoretical investigation section, a series of model CQDs consisting of different numbers of fused benzene rings were selected for calculation. The largest number of fused benzene rings of triangular model CQDs is 19, and the calculated emission peak (481 nm) and theoretical size (1.72 nm) are well consistent with that of the B-NBE-T-CQDs determined experimentally (emission peak: 472 nm, size: 1.9 nm). This clearly demonstrated that the DFT calculation results are appropriate to support our experimental results.

Reviewer 3

The Authors unfortunately did not take my comments seriously.

Reply: We did take the Reviewer's comments very seriously, but really apologize that we might have missed the Reviewer's main points of the comments. When we were emphasizing the achievement of the high color-purity narrow bandwidth emission (blue to red) of carbon quantum dots (CQDs), which was a breakthrough for CQDs in the display application of LEDs, we did not realize that some inappropriate descriptions might give a false sense that the concept of triangular graphene quantum dots was put forward by us for the first time. We sincerely appreciate for the Reviewer's reminder: "Otherwise, one must conclude that since the triangular graphene quantum dots with exactly known structure were already made, measured and theoretically analysed, there is not enough novelty to warrant publication in NatComm". Once again, we thank the Reviewer's high evaluation "a convincing demonstration of narrow emission with systematic size dependence of these graphene quantum dots would be really exciting" in the first round of review.

Therefore, we revised the manuscript carefully according to the Reviewer's suggestions, and sincerely hope that the Reviewer will recognize the significance of realizing high color-purity narrow bandwidth emission from CQDs as well as their applications for high color-purity and high-performance electroluminescent LEDs in our manuscript.

There are several papers, experimental and theoretical, which describe optical properties of triangular graphene quantum dots (TGQDs). These dots have exactly either 168 or 132 carbon atoms hence the size distribution is ZERO. The absorption and emission spectra were measured, including time resolved absorption, with one of most recent measurements reported in Sun, Figge et al. Nano Letters 15,5742, 2015). One can easily trace all experiments on these TGQDs from this paper in Nano Letters. The theory of optical properties of TGQDs was given in a paper by Ozfidan et al. Phys. Rev. B89,085310 (2014).

Reply: We agree with the Reviewer very much. There were several papers, experimental and theoretical, which described optical properties of triangular graphene quantum dots (T-GQDs). (Ozfidan et al. Phys. Rev. B 89, 085310 (2014)) set up two model triangular C168 and C132 GQDs, and their band-gap renormalization, electron-hole attraction, oscillator strength, and polarization of the exciton were theoretically investigated. In a following paper (Sun, Figge et al. Nano Letters 15,5472, 2015), the biexciton binding of Dirac fermions of triangular C168 GQDs was investigated with transient absorption measurements and microscopic theory although no any high color-purity narrow bandwidth emission was observed. The Authors decided to ignore these references. They continue to introduce GQDs using terms like "it is widely believed" and portraying the triangular shape of graphene dots as their major innovation. This is not correct.

Reply: We are very sorry that our inappropriate descriptions might give rise to the misunderstanding of ignoring these references. We mentioned "it has been widely accepted that CQDs give broad emission and inferior color-purity with full width at half maximum (FWHM) commonly exceeding 80 nm" to emphasize the significance of realizing high color-purity narrow bandwidth emission from CQDs as well as their applications for high color-purity and high-performance electroluminescent LEDs.

The Authors must start their paper by analysing state of knowledge, i.e., all existing experiments and theory of TGQDS with exactly known number of carbon atoms and their measured optical properties including excitonic and bi-excitonic states, and focus on convincing the reader what is the novelty of their work. Otherwise, one must conclude that since the triangular graphene quantum dots with exactly known structure were already made, measured and theoretically analysed, there is not enough novelty to warrant publication in Nat Comm.

Reply: According to the Reviewer's constructive suggestions, we have made corresponding changes in our manuscript as following:

1) In the introduction section, after discussing the great significance to realize high color-purity narrow bandwidth emission from CQDs, we immediately start a paragraph to discuss about the previous knowledge of triangular GQDs (T-GQDs) including theories and experiments. The detailed description of the added paragraph is as following:

"Indeed, the physical properties including band-gap renormalization, electron-hole attraction, oscillator strength, and exciton polarization of model triangular graphene quantum dots (T-GQDs) containing 168 and 132 sp^2 -hybridized C atoms have been theoretically investigated³⁴. Shortly thereafter, the biexciton binding of Dirac fermions of T-GQDs containing 168 sp^2 -hybridized C atoms has also been probed by transient absorption measurements and microscopic theory³⁵. "

In addition, the literatures of (Ozfidan et al. Phys. Rev. B 89, 085310 (2014)) and (Sun, Figge et al. Nano Letters 15, 5472, 2015) have been cited as reference 34 and 35 in the new added paragraph in the introduction section.

2) In the introduction section, we have removed the word "first" in the sentence "Herein, we report the first synthesis of high color-purity, narrow bandwidth (FWHM of 29 nm) and multicolored (from blue to red) emission triangular CQDs (T-CQDs) with a quantum yield up to 72%". The revised sentence is as following:

"Herein, we report the synthesis of high color-purity, narrow bandwidth (FWHM of 29 nm) and multicolored (from blue to red) emission triangular CQDs (T-CQDs) with a quantum yield up to 72%."

3) In the abstract section, we have removed the word "first" in the sentence "Here we first demonstrate high color-purity, narrow bandwidth (FWHM of 29-30 nm) and multicolored (from blue to red) emission from specially designed triangular CQDs (T-CQDs), with a quantum yield up to 54-72%". The revised sentence is as following:

"Here we demonstrate high color-purity, narrow bandwidth (FWHM of 29-30 nm) and multicolored (from blue to red) emission from specially designed triangular CQDs (T-CQDs), with a quantum yield up to 54-72%."

4) In the conclusion section, we removed the word "first" in the sentence "We report the first subversive demonstration of high color-purity NBE-T-CQDs (FWHM of 29-30 nm) from blue to red with a QY up to 54-72%". The revised sentence is as following:

"We report the subversive demonstration of high color-purity NBE-T-CQDs (FWHM of 29-30 nm) from blue to red with a QY up to 54-72%."

We thank the Reviewer again, and sincerely hope that the Reviewer would agree about the significance of realizing high color-purity narrow bandwidth emission from CQDs as well as their applications for high color-purity and high-performance electroluminescent LEDs in our manuscript, and find this paper now satisfactory for publication in Nature Communications.

Reviewers' Comments:

Reviewer #2:

Remarks to the Author:

The authors had properly responded my concerns, and I do not have other questions. So I would like to recommend this manuscript to be accepted for publication.

Reviewer #3:

Remarks to the Author:

The revised manuscript successfully addresses my concerns and is now appropriate for publication.